# Hydrostreamer v1.0 - improved streamflow predictions for local applications from an ensemble of downscaled global runoff products

Marko Kallio[1,2] *, Joseph H.A. Guillaume[3], Vili Virkki[2], Matti Kummu[2], Kirsi Virrantaus[1]

[1] Geoinformatics Research Group, Department of Built Environment, Aalto University, Espoo, Finland
5  [2] Water and Development Research Group, Department of Built Environment, Aalto University, Espoo, Finland
[3] Institute for Water Futures & Fenner School of Environment and Society, Australian National University, Canberra, Australia

*Correspondence to*: Marko Kallio (marko@markokallio.fi)

**Abstract.** An increasing number of different types of hydrological, land surface, and rainfall-runoff models exist to estimate streamflow in river networks. Results from various model runs from global to local scale are readily available online. However, the usability of these products is often limited, as they often come aggregated in spatial units which are not compatible with the desired analysis purpose. We present here an R package, a software library *hydrostreamer v1.0* which aims to improve the usability of existing runoff products by addressing the Modifiable Area Unit Problem, and allows non-experts with little knowledge of hydrology-specific modelling issues and methods to use them for their analyses. Hydrostreamer workflow includes 1) interpolation from source zones to target zones, 2) river routing, and 3) data assimilation via model averaging, given multiple input runoff and observation data. The software implements advanced areal interpolation methods and Area-to-Line interpolation not available in other products, and is the first R package to provide vector-based routing. Hydrostreamer is kept as simple as possible – intuitive with minimal data requirements – and minimizes need for calibration. We tested the performance of hydrostreamer by downscaling freely available coarse-resolution global runoff products from the Inter-Sectoral Impact Model Intercomparison Project (ISIMIP) in an application in 3S Basin in Southeast Asia. Results are compared to observed discharges as well as two benchmark streamflow data products, finding comparable or improved performance. Hydrostreamer v1.0 is open source and is available from http://github.com/mkkallio/hydrostreamer/ under MIT licence.

## 1    Introduction

An increasing number of different types of hydrological, land surface, and rainfall-runoff models exist to estimate streamflow in river networks. Alternative models with different assumptions, model structures and process representations increase the options to address specific hydrological problems. Using these models, however, requires skill and training to overcome barriers that inexperienced modellers or non-experts often face; for instance, hydrological jargon (Venhuizen et al., 2019), hydrological textbooks focusing on equations (Shaw et al., 2019), the curse of equifinality (Beven, 2006), selection of model performance indicators (Krause et al., 2005), selection of an appropriate model (Addor and Melsen, 2019; Singh and Woolhiser, 2002), and data collection, among others (Brunner et al., 2021). A non-expert (by which we mean a person who is interested in hydrological information but is not an expert in hydrological modelling) has several options to choose from when facing this problem. They can, for instance, seek help of an expert hydrologist, possibly incurring costs to their project, or delayed delivery if parts of the project need to wait for input from the hydrologist. Or they can apply a model with simplified process representations or which are designed for teaching. These models include e.g. a simple rainfall-runoff ratio, Khosla's Method (Subramanya, 2017), HBV (Seibert and Vis, 2012), or airGR (Delaigue et al., 2018), all of which still require data collection, and calibration or obtaining parameter estimates from an external source. Using a model evaluation framework (e.g. Hamilton et al. 2019), an inexperienced hydrological modeller may in this case encounter

challenges relating to impacts at project level (efficiency, credibility, salience, accessibility) and group level (application and satisfaction).

Alternatively, the non-expert can use readily available hydrological data products prepared by expert teams, sidestepping many of the aforementioned issues. The drawback of using off-the-shelf existing products is that they often come aggregated in spatial units which are not compatible with the desired analysis purpose. This issue is termed as the Modifiable Area Unit Problem (MAUP) - a statistical bias in analyses arising from arbitrarily defined aggregation zones, resulting in both a scale effect and zoning effect (Manley, 2014). The scale effect occurs when the aggregation of a statistic to different scale enumeration areas (such as catchment, basin or watershed) produce different statistics. The zoning effect occurs when different arrangements of enumeration areas (such as a regular grid, and a polygon type administrative area) produce different values for the same sample location. If MAUP is not addressed during an analysis, the analyst runs a risk that the data used is not representative of their units of analysis. Salmivaara et al. (2015) explore MAUP for water resources assessments in more detail.

A number of options to address MAUP have been developed (Dark and Bram, 2007), but no comprehensive solution has been found. Solutions range from simply ignoring MAUP to analysing each elementary areal unit relevant for the variable or process. However, ignoring the problem and hoping for the best does not seem like a desirable solution, and analysis of every possible elementary unit may not be computationally feasible. One of the solutions is to use some areal interpolation method to estimate variable values in alternate aggregation zones (Kar and Hodgson, 2012). Such areal interpolation methods include any method which estimates an unknown variable value in a *target* zone based on known values in a *source* zone (Goodchild and Lam, 1980). In the context of hydrological modelling, a considerable body of literature exists for statistical interpolation of hydrological variables to ungauged basins (e.g. Gottschalk, 1993; Lehner and Grill, 2013; Paiva et al., 2015; Parajka et al., 2015; Skøien et al., 2006). Process-based interpolation has gained little attention outside our previous work (Kallio et al. 2019) and that of Kar and Hodgson (2012), who propose a method of incorporating process-understanding in application of advanced areal interpolation methods for downscaling runoff, and for estimating population densities, respectively.

Notwithstanding the issues with MAUP, the workflow of estimating discharge from runoff remains similar. Regardless of the method used to estimate runoff, once the quantity of runoff is approximated for a certain area, discharge is commonly derived by applying a river routing algorithm which accumulates the runoff down a river network. Distributed and semi-distributed hydrological models often have a built-in routing component. Nevertheless, hydrological variables are also modelled by land-surface models and dynamic vegetation models, which may require coupling with a routing model. A number of software solutions exist which first interpolate runoff to arbitrary river reaches and apply various routing methods, either on a node-link network or between grid cells representing the river system. Focusing here on node-link networks, examples of such tools are Routing Application for Parallel computatIon of Discharge (RAPID; David et al., 2011),

mizuRoute (Mizukami et al., 2016), and HYDROROUT (Lehner and Grill, 2013). While these tools do include a step to map runoff products onto the river network, their focus is primarily on routing (RAPID and MizuRoute are high-performance solutions), with interpolation limited to centroids and simple area weighted interpolation.

Furthermore, many authors have recognised the need for multiple estimates for robust problem solving in hydrology (e.g.
Addor and Melsen, 2019; Blair and Buytaert, 2016) using alternative model structures, assumptions and source datasets. This is currently not explicitly handled in available interpolation and routing tools. The inclusion of several estimates is commonly achieved through uncertainty quantification (Vrugt and Robinson, 2007), but when using existing source datasets, these typically represent a sample of convenience, and it may be of limited use to focus on uncertainty across interpolation methods. Instead, it makes sense to treat the use of multiple estimates as a model selection problem, or more generally, a
model averaging (MA) problem, where predictive performance dictates which methods to select, or their contribution to a combined prediction (Diks and Vrugt, 2010). Performance is expected to vary spatially and temporally and observation data is not available for all cases, so methods are then also needed to select models in areas without performance information. MA can additionally contribute to solving MAUP, where estimates from alternative zonings are used as inputs for MA. Such alternative zonings in the context of water resources assessments may be different realizations of uncertain drainage basin
delineations (Eränen et al., 2014) or combinations of physical and administrative delineations (Salmivaara et al., 2015).

This paper responds to the need for downscaling of existing runoff products in a way that addresses the MAUP, links to routing functionality, and allows for use of model averaging on an ensemble of convenience. Further, the use of existing runoff products relaxes the requirements for hydrological modelling expertise from the user's part, provided that they possess the expertise in R programming and elementary skills in working with spatial data. The software introduced presents
an alternative to existing software tools to provide further options to tackle the scale effect (downscaling, upscaling) and zoning effect (similar resolution, but non-conforming zones) of the MAUP. *Hydrostreamer v1.0*, a software library written in the R language (R Core Team, 2019), uniquely combines 1) advanced areal interpolation methods, 2) lateral routing methods, and 3) functions for data assimilation via a model averaging approach in order to improve and extend the usability of off-the-shelf runoff products. Hydrostreamer particularly implements advanced areal interpolation methods, 1) dasymetric
mapping (DM), areal interpolation which is guided by ancillary variables (Comber and Zeng, 2019; Eicher and Brewer, 2001; Wright, 1936), and 2) pycnophylactic interpolation (PP), a technique to estimate internal variable distribution within a specified zone (Tobler, 1979). To our knowledge, there are currently no other software solutions for R which implement DM, but area weighted interpolation is available in at least *sf* (Pebesma, 2018) and *areal* (Prener and Revord, 2019) R packages. Tobler designed PP for an internal representation using a square grid scheme, and Rase (2001) developed an
adaptation of PP for Triangulated Irregular Networks. To the best of our knowledge, hydrostreamer is the first software package which implements PP for polygon networks as the internal representation, but a gridded version is available for R in package *pycno* (Brunsdon, 2014). Hydrostreamer also implements an Area-to-Line interpolation which supports using river

networks obtained from mapping surveys or topographic databases as an alternative to networks extracted from Digital Elevation Models, which can be highly uncertain (Lindsay and Evans, 2008).

We demonstrate the capabilities of hydrostreamer with a case study in the data-poor 3S basin in Southeast Asia. The 3S basin is a major tributary of the Mekong River, consisting of three rivers Sekong, Sesan and Srepok under Southwest

Monsoon climate. In the case study, we use hydrostreamer to downscale 15 runoff products obtained from the Inter-Sectoral Impact Model Intercomparison Project (ISIMIP; simulation experiment 2a (Gosling et al., 2017)) on to HydroSHEDS river network (Lehner et al., 2008). Following this, we route the downscaled runoff down the river network and perform model averaging against streamflow records from 10 monitoring stations. Performance of streamflow predictions against gauged observations is compared with benchmarks of a recent streamflow product, GRADES (Global Reach-Level A Priori

Discharge Estimates for SWOT; Lin et al., 2019) and ECMWF published global streamflow reanalysis product, GLOFAS (GLObal Flood Awareness System; Alfieri et al., 2020).

The rest of the paper is structured as follows. In Sect. 2 we introduce the four steps of preprocessing, areal interpolation, routing, and model averaging in hydrostreamer. In Sect. 3 we introduce the software architecture. Sect. 4 describes the case study method, data and experiments, for which the results and discussion are provided in Sect. 5. Sect. 6 discusses future

development plans, and Sect. 7 gives conclusions.

## 2    Core hydrostreamer v1.0 functionality

*Hydrostreamer* is designed as a complete solution from preprocessing source data to interpolation, river routing, post-processing outputs and evaluating model performance. It is written in the R language (R Core Team, 2019), which is receiving increasing attention in the hydrological sciences (Slater et al., 2019). We have aimed to keep the core functionality

of the package as simple as possible to facilitate its use for non-experts while implementing enough functionality to be useful also for hydrological community. Figure 1B shows the generalized steps taken in a typical hydrostreamer workflow. This section is structured as follows: Obtaining data and preprocessing are discussed in Sect. 2.1, interpolation methods (Step I) are described in Sect. 2.2, river routing (Step II) in Sect. 2.3, and data assimilation by model averaging (Step III) in Sect. 2.4.

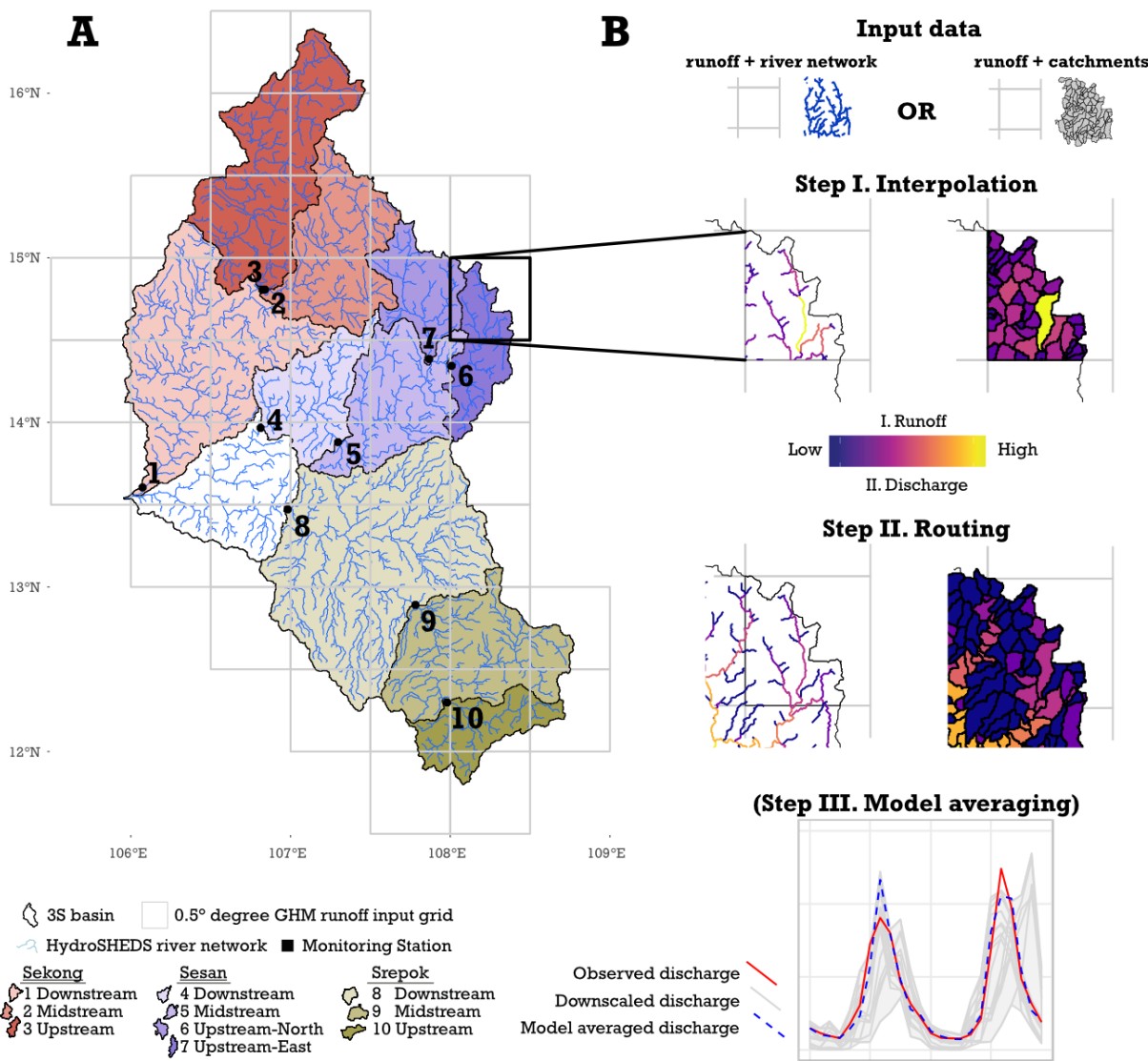

Figure 1. A) Study area and the data used in the empirical study: runoff from global hydrological models at 0.5° resolution, a river network, and monitoring data. B) The three steps in a typical hydrostreamer workflow. Step I: areal interpolation, or Area-to-Line interpolation, to distribute runoff from source zone to target zones (or river lines). Step II: river routing. Step III: model averaging to create a Multi-Model Combination (and regionalisation) if data from monitoring stations is available.

## 2.1 Obtaining data and preprocessing

Hydrostreamer has been designed to work with very low data requirements consisting of, at minimum, a distributed runoff dataset as the source zones, and an explicit river network as the target zones, both of which can be obtained from free and open repositories. Additional data in the form of a Digital Elevation Model (DEM), or ancillary information on the target river network can be used to further improve the streamflow estimates.

Hydrostreamer is designed to take runoff in a raster timeseries format – a multi-layer raster where each layer corresponds to a specific timestep – as a plug-and-play solution. Such datasets include, for instance, LORA (Linear Optimal Runoff Aggregate; Hobeichi et al., 2019) and GRUN (Ghiggi et al., 2019), both of which are optimised global runoff datasets at 0.5° degree resolution. Outputs of a large number of global hydrological and land-surface models can also be obtained from the ISIMIP archive (as is done in the case study described in Section 4; https://www.isimip.org/) at the same 0.5° resolution. In order to use this type of data in hydrostreamer, one needs to read it in the R session, and use *raster_to_HS()*-function to format the data for hydrostreamer to use. Non-gridded data may also be used, and one potential source dataset is GRADES (Lin et al., 2019), which is used as benchmark dataset in this study. While hydrostreamer supports non-gridded input runoff via the function *create_HS()*, their use may not be trivial, and depending on the source format, may require additional preprocessing steps.

The river network required by hydrostreamer can be obtained from various sources. Any network which is topologically clean can be used as a plug-and-play solution without any preprocessing needed when Step I (Figures 1 and 2) is applied. A clean network refers to a network where connected lines share a node and lines are split at junctions so that connected line segments start or end at a shared node. With a clean network, *river_network()* function is able to extract the topological relationships hydrostreamer needs. Further, if the used river network product contains topological information on the *next* river segment, the cleanliness requirement can be relaxed. The case study presented in Sect. 4 uses HydroSHEDS river network (Lehner and Grill, 2013), available from https://www.hydrosheds.org/, which is an example of a clean river network. The HydroSHEDS website provides additional river network datasets with a large number of attribute information describing each segment: Global River Classification (Dallaire et al., 2019) and HydroATLAS (Linke et al., 2019). These two datasets are particularly useful in hydrostreamer, because their attribute information can be (but is not required) used as ancillary variables in dasymetric mapping (DM,see the following Section 2.2).

Hydrostreamer further provides an optional auxiliary function *create_river()* which can be used to extract a river network and catchment areas from a DEM for each river segment. The function requires an external program, SAGA GIS (Conrad et al., 2015) to be installed, and requires definition of a threshold for the size of the stream (the Strahler stream order) at which point river line extraction starts. The selection of the threshold should be guided by the resolution of the source zones as well as understanding of the hydrology within the basin. We recommend visual inspection of the extracted river network as well as their corresponding catchment areas. Catchment areas can also be delineated  for each individual river segment from a

flow direction raster using function *delineate_basin()*, if the flow direction information from which the river network is derived is available. For HydroSHEDS, HydroATLAS and Global River Classification, the flow direction information can be obtained from the same repository and can be used with the *delineate_basin()* function. Further, when an applicable DEM or flow direction are not available, function *river_voronoi()* approximates catchment areas by building a Thiessen polygon

(Voronoi diagram) network from the line segments of the river network (Karimipour et al., 2013). This is particularly useful for river networks derived from surveying or from satellite measurements.

Finally, in order to use the model averaging functionality (optional, Step III in Figure 1B and Figure 2) of hydrostreamer, one needs to obtain a timeseries of discharge measurements at gauges corresponding to the catchment of interest. We recommend obtaining timeseries from the authorities of administrative area(s) where the catchment is located, but when this

is not possible, one can search e.g. Global Streamflow and Metadata archive (Do et al., 2018) or the Global Runoff Data Centre (https://www.bafg.de/GRDC/EN/Home/homepage_node.html) for appropriate data. Hydrostreamer requires observation data in a standard table format with columns for *Date*, and station observations in units of $m^3 s^{-1}$.

## 2.2    Step I: Areal interpolation in Hydrostreamer

The first step in the core hydrostreamer workflow is the areal interpolation step, which is also the key focus of the software

package. In this section we give a brief background for the areal interpolation methods and how they are implemented in hydrostreamer. For a more thorough overview and applications of areal interpolation methods, we recommend Comber and Zeng (2019). The current implementation in hydrostreamer assumes that the interpolation is constant and does not change through time.

### 2.2.1    Area-weighted interpolation and dasymetric mapping

Areal interpolation methods have been developed in geography to represent regionally aggregated statistics in non-conforming area units, and are discussed mostly in literature for population mapping (Comber and Zeng, 2019; Eicher and Brewer, 2001; Goodchild et al., 1993; Goodchild and Lam, 1980; Nagle et al., 2014; Wright, 1936). In principle, areal interpolation involves reallocation of a quantity from a source zone to intersecting target zones. The simplest form of areal

interpolation is the area-weighted interpolation (AWI), where the reallocation is based on the proportion of intersecting areas, as shown in Eq. (1),

$$\widehat{RO}_t = \sum_s^{s \cap t} RO_s \frac{A_{t \cap s}}{A_s} \tag{1}$$

where $\widehat{RO}_t$ is the estimated value of the variable of interest in a target zone $t$, $RO_s$ is the value in a source zone $s$, $A_s$ is the area of the source zone, and $A_{t \cap s}$ is the area of the intersection of the target zone with the source zone. This form of areal interpolation is a standard practise in many hydrological applications.

The reallocation can, however, be guided by ancillary variables in dasymetric mapping (DM) if we know the process behind the interpolated variable, and additional variables describing the process are available. "Dasymetric" means *density measuring*, and DM is sometimes referred to as "intelligent areal interpolation" (Eicher and Brewer, 2001). In DM, the areal weights derived from AWI are further scaled using the values of ancillary variable. With the added ancillary variable $V_t$, Eq. (1) becomes

$$\widehat{RO}_t = \sum_s^{s \cap t} RO_s \frac{A_{t \cap s} V_t}{\sum_t^{t \cap s} A_{t \cap s} V_t} \tag{2}$$

where $V_t$ is the value of the ancillary variable for the target zones and $A_{s \cap t}$ is the area of source zone intersecting all target zones. $V_t$ can be any (numerical) variable which describes the distribution of the interpolated quantity within target zones. By definition, both AWI and DM are volume or mass preserving (pycnophylactic) – the quantity of the interpolated variable from a source zone is divided exactly among intersecting target zones. The variable $V_t$ can also be substituted with a model $f(RO)$ describing the process behind the variable being interpolated. This is called dasymetric *modelling* – see e.g. Kar and Hodgson (2012) or Nagle et al. (2014). The dasymetric variable(s) should be selected such that it (they) describe the distribution of runoff *within* each source zone. Potential variables include topographic information (elevation, topographic indices; the case study in this paper uses a topographic index as a dasymetric variable), landuse, soil type, climate information (precipitation, temperature, evapotranspiration), and so on. The choice depends on the availability of data for each individual target zone as well as on the hydrological understanding of the user.

### 2.2.2   Area-to-Line interpolation

AWI and DM both require that the target zone is reliably delineated with no significant uncertainty. To avoid this requirement, hydrostreamer also provides both methods adapted for line-features, which we call Area-to-Line interpolation. This is achieved by replacing a target zone's area $A$ with target line's length $L$. In the context of river networks, both area and length are physical attributes of the river segment; one describing the catchment area associated with the river line and the other describing the river line itself. With this modification, Eq. (1) becomes

$$\widehat{RO}_l = \sum_s^{s \cap l} RO_s \frac{L_{t \cap s}}{\sum_l^{l \cap s} L_{t \cap s}} \tag{3}$$

where $\widehat{RO}_l$ is the estimated value of the variable of interest in a target line $l$, and $L_{l \cap s}$ is length of the intersecting portion of river line $l$ within the source zone. Similarly, Eq. (2) becomes Eq. (4):

$$\widehat{RO}_l = \sum_s^{s \cap l} RO_s \frac{L_{l \cap s} V_l}{\sum_l^{l \cap s} L_{l \cap s} V_l} \tag{4}$$

In some combinations of river lines and source zones, the river may flow exactly along the boundary of two or more source
zones. Since this portion intersects both source zones, such cases are explicitly handled by hydrostreamer to split the contribution evenly among the source zones for the portion of river line at the boundary.

While the computation is similar, and both areal interpolation and Area-to-Line are pycnophylactic, areal interpolation can by definition work with a partial overlap between source and target zones (river segment lines). For Area-to-Line interpolation, there is no area representation of the target area, and therefore the used river network must intersect all source
zones and should be represented in similar accuracy throughout each source zone. In our case study presented in Sect. 4, each source zone (a 0.5° grid; approximately 55 km at the equator) intersects on average 56 river segments. As the performance difference is small between Area-to-Line interpolation and area-based interpolation methods (Appendix A, Table A1), this can be considered a sufficient density for source zones in this resolution (but subject to case-by-case evaluation). Further, since the individual river segment length is not directly proportional to its individual catchment area,
Area-to-Line interpolation should only be used for sufficiently large basins, where the area of source zones entirely contained in the basin is significantly larger than the area of partially covered source zones. Based on our case study, monitoring stations with a drainage area of at least 30 000 km$^2$ show very small performance difference between Area-to-Line interpolation and area-based interpolation methods (Table A1). Due to the large uncertainty in runoff distribution to individual segments, we recommend that the suitability of Area-to-Line interpolation be performance-evaluated on a case-
by-case basis.

### 2.2.3    Pycnophylactic interpolation

While other interpolation methods may also be volume or mass preserving, pycnophylactic interpolation (PP) refers to a class of methods developed by Tobler (1979) to estimate the internal variation of a variable within a certain source zone.
Tobler's application first subdivides a source zone into a regular grid and creates a smooth representation of the interpolated variable which preserves the volume within the source zone. The smoothing involves solving an integral in both x- and y-direction, which is subject to the condition in Eq. (5) that preserves total volume $RO_s$ across the parts of target zones $t$ within the original zone $s$. (Kar and Hodgson, 2012; Tobler, 1979)

$$\sum_t^{t \cap s} \widehat{RO}_t = RO_s \tag{5}$$

Rase (2001) developed an adaption of PP which uses Triangulated Irregular Networks (TINs), where the double integral is simplified to averaging over nearest neighbours and weighting neighbours with Inverse Distance Weighting. Hydrostreamer implements PP for polygon networks by adapting Rase's approach to the immediate neighbours of each target zone. This is achieved by iteratively alternating an averaging step and an adjustment step to satisfy the condition in Eq. (5). Adapting the approach of Rase, we get an averaging step

$$\widehat{RO}_t = \frac{RO_t + \sum_j^N RO_j}{N+1} \tag{6}$$

where $N$ is the number of neighbours *adjacent to* target zone $t$, and $RO_j$ is the value of neighbour $j$. The averaging step is followed by an adjustment step, where the target zones within a source zone are scaled so that Eq. 5 condition is met. If a target zone is at the boundary of the area of interest, the boundary condition is set as the starting value of the target zone at the beginning of PP. The boundary condition does not change with iterations, consistent with the suggestion of Tobler (1979). It should be noted that averaging over neighbours is done for *density* (i.e. runoff depth), but the smoothing condition is applied for *volume* (i.e. runoff volume across source zones) in order to satisfy the pycnophylactic property of PP.

PP cannot be used with river line features due to lack of computable area, and non-trivial measures of neighbours. However, PP can be applied if drainage areas are estimated for the river network, for example using river segment-specific Thiessen polygons (Karimipour et al., 2013). Thiessen polygons for a line network can be computed with hydrostreamer.

### 2.2.4 Combined pycnophylactic-dasymetric interpolation

Hydrostreamer implements a possibility to utilize a combination of PP and DM as described in Kallio et al. (2019). In the combined PP-DM method, the initial density of an interpolated variable for each target zone is first estimated with PP (instead of AWI), followed by DM. In this version, Eq. (2) becomes

$$\widehat{RO}_t = \sum_s^{s \cap t} RO_s A_{s \cap t} \frac{RO_{pp,t} A_{t \cap s} V_t}{\sum_t^{t \cap s} RO_{pp,t} A_{t \cap s} V_t} \tag{7}$$

where $RO_{pp,t}$ is the value of $RO$ for target zone $t$, as initially estimated by PP. The advantage of the combination is that through PP we can model variables which are assumed smooth (e.g. precipitation) within and between source zones, and DM is used to estimate crisp processes.

## 2.3    Step II: River routing

The second step in a typical hydrostreamer workflow involves routing runoff down a river network to estimate discharge. Two simple routing solutions, instantaneous routing and constant-velocity routing, as well as one more advanced routing solution, the Muskingum-Cunge method, are implemented in hydrostreamer. More sophisticated schemes could be used by exporting interpolated runoff to other tools specialising in routing.

In instantaneous routing, discharge $Q$ is the sum of runoff, in volume per time (e.g. m$^3$ s$^{-1}$), from all upstream catchments, as shown in Eq. (8):

$$Q_{i,t} = RO_{i,t} + \sum RO_{up,t} \tag{8}$$

Here $Q_{i,\mathrm{t}}$ is the discharge at river segment $i$ at timestep $t$, $RO_{i,t}$ is the runoff contribution from the catchment area of segment $i$ at the same timestep $t$, and $RO_{up,t}$ is the runoff from a river segment upstream of segment $i$, at timestep $t$. Instantaneous flow is the simplest form of river routing and has the advantage that it is intuitive. However, it assumes that all runoff generated at a timestep $t$ will drain through the entire river network within that same timestep. The applicability of this assumption is therefore limited to catchments where the timestep length far exceeds the maximum river network length. One can evaluate the applicability of the instantaneous routing (which in fact takes one timestep) using Eq. (9):

$$M = \frac{\max L_{up}}{V_{max}} \frac{1}{s} \tag{9}$$

where $M$ is a dimensionless ratio between the time it takes for water to flow through the maximum upstream length of the river system $L_{up}$ (in meters) at a maximum realistic average flow velocity $V_{max}$ (default 1 meter per second) during a timestep of length $s$ (in seconds). $M$ can be interpreted so that, for example, when $M = 0.1$, 10% of the runoff generated at the most distant upstream location does not flow through the outlet within a single timestep. The evaluation can be carried out using the function *evaluate_instant_routing()*. If $M$ is found to be too large for the application, the constant flow velocity or Muskingum-Cunge option may be more appropriate.

The second option, constant velocity routing, assumes that water drains through the river network at a constant pace, which is the solution adopted also in routing tool HydroROUT (Lehner and Grill, 2013) and a number of GHMs (Telteu et al., 2021). The default flow velocity of 1 meter per second is adopted e.g. in HydroROUT and LPJmL (Telteu et al., 2021). Assuming that the generation of runoff is uniformly distributed within a timestep, we can think of a block of runoff moving downstream. Given a distance $S$ covered in time $\tau$ at velocity $V$, the block of runoff typically finds itself spanning several river segments (depending on the segment size and timestep length). The discharge in a particular segment comes from two

blocks of runoff (two consecutive timesteps) from each upstream segment. We calculate the number of whole timesteps $T_{i,up}^{whole}$ elapsed for the runoff to cover the distance $D_{i,up}$, and the fractions $T_{i,up}^{-}$ that will come from that timestep and $T_{i,up}^{+}$ from the following timestep. See Appendix B, Figure B1. The process is described in Eq. (10-14):

$$S = \frac{V}{\tau} \tag{10}$$

$$T_{i,up}^{whole} = floor\left(\frac{D_{i,up}}{S}\right) \tag{11}$$

$$T_{i,up}^{+} = \text{frac}\left(\frac{D_{i,up}}{S}\right) \tag{12}$$

$$T_{i,up}^{-} = 1\text{-frac}\left(\frac{D_{i,up}}{S}\right) \tag{13}$$

$$Q_{i,t} = \sum_{up}^{upstream} RO_{up,T_{i,up}^{whole}} \times T_{i,up}^{-} + RO_{up,T_{i,up}^{whole}+1} \times T_{i,up}^{+} \tag{14}$$

The third routing option implemented in hydrostreamer is the Muskingum-Cunge routing algorithm (Cunge, 1969; Ponce, 2014). Muskingum-Cunge is a modified version of the original Muskingum routing method (Chow, 1959) where routing parameters $k$ and $x$ are derived from hydraulic data and does not require observation data to calibrate against. Full derivation and explanation of the Muskingum-Cunge routing can be found in Ponce (2014). The algorithm requires extensive user input in the form of river cross-sections (i.e. shape, channel width, flow depth), river bed roughness (Manning's roughness coefficient), and river bed slope, which are commonly available only for certain locations. Consistent with the desire to minimise data requirements, the hydrostreamer implementation of Muskingum-Cunge provides defaults and therefore requires the user only to provide main parameters: 1) Manning's roughness coefficient (for readers unfamiliar with Manning's coefficient, Arcement and Schneider (1989) provide an extensive guide on its estimation), 2) bed slope (precomputed bed slopes are available e.g. from the HydroATLAS (Linke et al., 2019) database which can be directly used in hydrostreamer), and 3) channel width. An estimate of the channel width can be computed using a power-law relationship (Leopold and Maddock Jr., 1953)

$$W = aQ_{ref}^{b} \tag{15}$$

where $a$ and $b$ are parameters to be estimated and $Q_{ref}$ is the reference discharge, and $W$ is the channel width. Hydrostreamer has a built-in estimates for $a$, $b$ from Moody and Troutman (2002) and Allen et al. (1994). $Q_{ref}$ is estimated from the inflowing discharge timeseries for each river segment using Eq. 16,

$$Q_{ref} = \min(Q_{in}) + \frac{\max(Q_{in})-\min(Q_{in})}{2} \tag{16}$$

where $Q_{in}$ is the timeseries of discharge inflowing to the river segment. Alternatively, the user can provide their own parameters for each $a$, $b$, and $Q_{ref}$ Vatankhah and Easa (2013) derived a relationship between discharge $Q$ and flow area based on channel width. Their approach is used here to estimate flow depth assuming a rectangular river cross-section.

All three routing solutions support setting boundary conditions which modify $RO$ at specified river segments. The boundary conditions, termed *control* timeseries, include addition, subtraction, multiplication, and setting $RO$ to a user-specified value. This allows inclusion of, for example, controlled inflows, water extraction, simple fractional environmental flow considerations, and specified dam releases, respectively.

For beginners, use of coarse timescales and instantaneous routing is recommended, subject to evaluation of performance. This will avoid the difficulties in estimating the parameters required for more complex algorithms. If additional information or expertise is available, the more complex routing algorithms may be selected to further improve performance or allow discharge estimation at shorter timescales. We provide a comparison of the three routing methods for the case study presented in Sect. 4 in Appendix B.

Note that, our case study example and Appendix B provide validation for the routing with monthly timeseries only. We therefore recommend caution and careful review of hydrostreamer outputs in applications using sub-monthly timeseries, until proper validation for the method is published.

## 2.4    Step III: Multi-model combinations

The third step in the typical hydrostreamer workflow is model averaging. Model averaging using varying sizes of ensembles is a common approach to data assimilation in hydrological sciences (see e.g. Arsenault et al. 2015; Arsenault and Brissette 2016; Gosling et al. 2010; Skøien et al. 2016; Velázquez et al. 2011; Zaherpour et al. 2019). In model averaging, an ensemble of timeseries is combined into a Multi-Model Combination (MMC), commonly using a weighted approach. Provided that streamflow records are available, hydrostreamer provides facilities for at-location-model averaging, as well as regionalised model averaging. Table 1 provides an overview of all the implemented model averaging methods in hydrostreamer along with some of their properties. The methods are based on minimising error between observations and the weighted ensemble average.

In a regionalisation experiment Arsenault and Brissette (2016) explore MMC weights using a small 3-member ensemble, finding that MMCs are nearly always outperformed by the best-performing individual model member at regionalized locations, and that simple ensemble mean (each ensemble member receiving equal weights) performs reasonably well across all locations. They further conclude that regionalizing model averaging weights is not a reasonable task, however their

conclusions are based on a limited analysis of three methods which all allow negative weights. We argue, that their conclusion applies to methods that allow for negative weights due to the fact that relationships between hydrological timeseries at different locations likely differ considerably. The lack of negative weights, we assume, is one reason that the ensemble mean is able to outperform their selected model averaging methods.

5    To help extend model averaging to ungauged basins (that is, any river segment which does not contain a monitoring station), we must find a way to regionalize model averaging weights. In hydrostreamer, we consider those methods which do not allow negative weights as fit for regionalization. This ensures that, while the performance of regionalized MMC weights may be worse than the best individual ensemble member (which we cannot know in an ungauged basin), the output hydrological timeseries is ensured to be positive.

10   In practise, in hydrostreamer regionalization of MMC weights is done so that each river segment receives model averaging weights from the nearest downstream gauging station, and if there are no downstream gauging stations, the ensemble mean is used instead.

**Table 1. Multi-Model Combination options implemented in hydrostreamer v1.0**

| Combination type | Abbreviation in hydrostreamer | Bias correction? | Weights sum to unity? | Allows negative weights? | Fit for regionalisation? | Notes |
|---|---|---|---|---|---|---|
| Constrained Least Squares | CLS | No | Yes | No | Yes | Assumes that observations are within the envelope of ensemble members |
| Non-negative Least Squares | NNLS | Implicit through weights | No | No | Yes | Bias not fully compensated |
| Granger-Ramanathan Type A | GRA | Implicit through weights | No | Yes | No | Bias not fully compensated |
| Granger-Ramanathan Type B | GRB | No | Yes | Yes | No | |
| Granger-Ramanathan Type C (Ordinary Least Squares) | GRC / OLS | Constant | No | Yes | No | Unbiased |
| Bates-Granger | BG | No | Yes | No | Yes | |
| Inverse Rank | InvW | No | Yes | No | Yes | |
| Standard Eigenvector | EIG1 | No | Yes | Yes | No | Bias not fully compensated |
| Bias corrected Eigenvector | EIG2 | Constant | Yes | Yes | No | Unbiased |
| Best | best | No | Yes | No | Yes | Picks the best individual ensemble member |

| | | | | | | User can provide a custom objective function which will be passed to *optim()* function of |
|---|---|---|---|---|---|---|
| User defined function | *Function* | - | - | - | - | the *stats*-package. |

## 3 Hydrostreamer v1.0 software architecture

Here we describe the Hydrostreamer workflow in Sect. 3.1 together with auxiliary functionality which help in making use of alternative data inputs and working with the output. Following that we describe the input and output data structures in Sect. 3.2.

### 3.1 Functions used in Hydrostreamer v1.0 workflow

The functions used in a typical workflow and data requirements in hydrostreamer are shown in the flow diagram in Figure 2. Due to the large number of optional arguments and functionality, the figure only shows minimum required inputs. Complete up-to-date documentation and default values can be found in the model documentation website at

, or using the internal R *help( )*-function. The documentation site also provides a vignette which gives further information on the workflow of hydrostreamer in the form of a practical example.

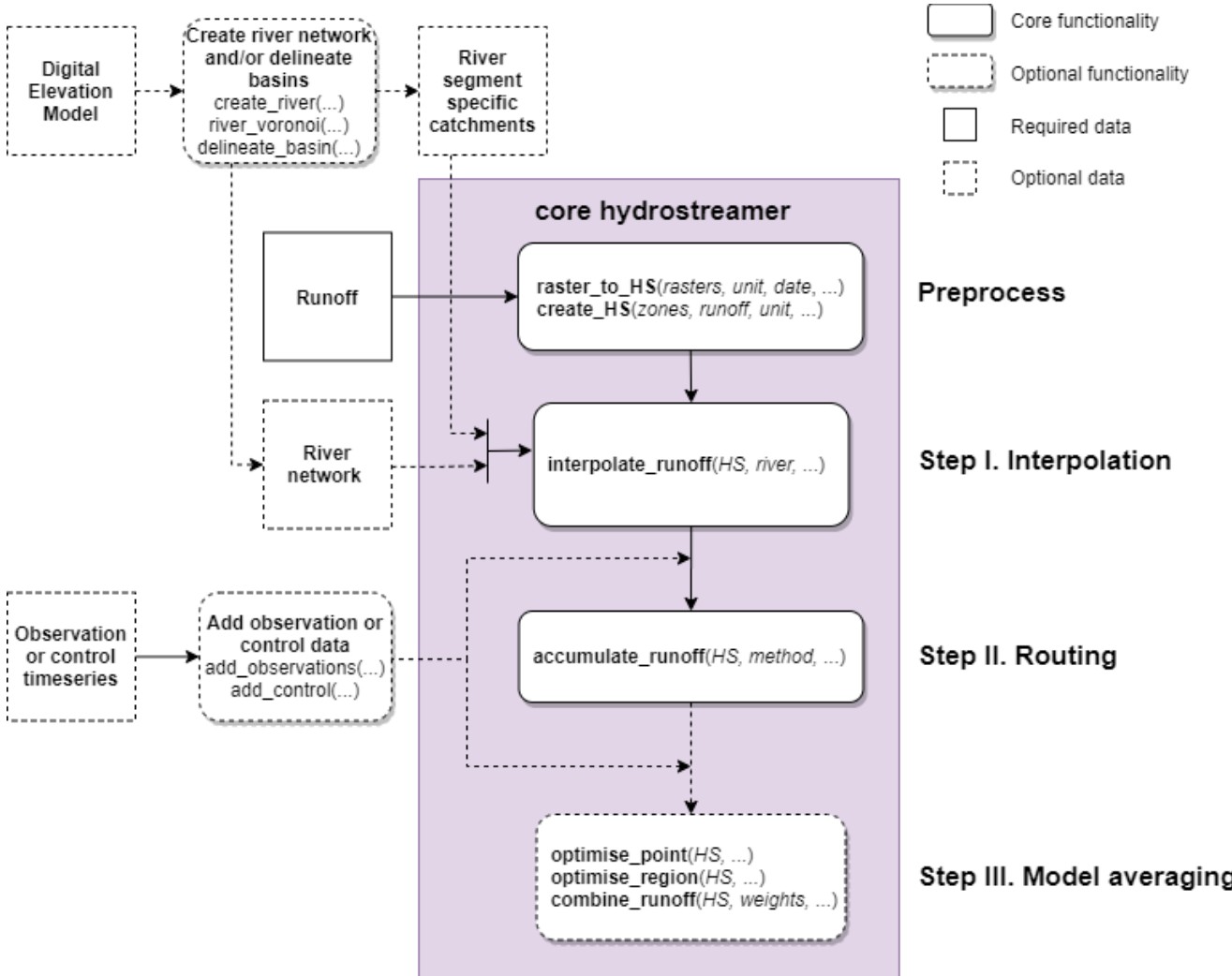

5    **Figure 2. Data requirements and related core workflow showing function names and minimal required input arguments. The steps refer to Figure 1. Ellipsis marks optional function arguments.**

The workflow starts from pre-processing data to a format with which hydrostreamer is able to work. Hydrostreamer supports providing input hydrological variables either as raster or vector formats, each with dedicated functions *raster_to_HS( )*, and *create_HS( )*. Each raster cell, or each polygon in the input data is considered as a source zone. For the interpolation step, a

10   network of target river lines and/or target zones need to be provided in addition to the *HS* object output from the pre-processing steps. The routing step requires the output from the interpolation step, the routing method (instantaneous, or

constant velocity), and parameters for that routing method. Finally, model averaging can be performed 1) at point location of the monitoring station, giving optimal timeseries for the monitoring station locations only, 2) regionalized model averaging, where MMC are provided for each river segments, optimized at the nearest downstream monitoring station, and 3) user provided combination weights.

5  Once runoff data and river network has been read into R, the full workflow can be achieved with only 3-5 chained commands, depending on whether the user wishes to apply model averaging. Hydrostreamer provides some additional functionality which support the optional components in Figure 2. These and further supporting functions are given in Table 2.

**Table 2. Auxiliary functions in hydrostreamer v1.0, their intended utility and their data and software requirements.**

| Function | Utility | Requirements |
|---|---|---|
| **create_river** | Derives a river network and catchment areas for each individual river segment from a provided DEM. | Requires *SAGA GIS* (Conrad et al., 2015) installed in the system. |
| **delineate_basin** | Delineates catchment areas for provided river segments from a flow direction raster. | Flow direction raster from which the river network is also derived. |
| **river_voronoi** | Derives an estimate of catchment areas by constructing Thiessen Polygons (Voronoi Diagram) from the river lines (Karimipour et al., 2013). Intended to enable areal interpolation methods for river networks, when Area-to-Line interpolation is unreasonable, and to enable use of PP for line networks. | |
| **river_hierarchy** | Compute Strahler stream order for the provided river network | |
| **river_network** | Derives topological information (next and previous river segments) for all river segments in the network and formats them for hydrostreamer. | Requires either already known topological information (next and/or previous segment), or a clean river network. In a clean network, intersections between river lines have a common node, and all lines are broken at the intersection. |
| **upstream, downstream** | Extract all downstream/upstream river segments from the network from a specified segment. | |
| **flow_gof** | Computes 20 goodness-of-fit measures commonly used in hydrology computed for all monitoring stations and all timeseries. | *hydroGOF* package (Mauricio Zambrano-Bigiarini, 2017) |
| **discharge, runoff, observation, control** | Convenience functions to extract the timeseries for a specified segment. | |

| compute_upstream_aggregate | Function to compute an aggregate of some variable from values recovered from all upstream segments. |
| --- | --- |
| compute_hydrological_signatures | Apply a user-provided function to for a timeseries (runoff, discharge, observation, control) column in *HS*-object. |
| evaluate_instant_routing | Function to help evaluate whether instantaneous routing can be used for a specific river basin. |
| compute_network_length | Computes the maximum length of upstream segments in the network |

## 3.2    Data structures

The data structures used in hydrostreamer are compatible with the packages from the *tidyverse* (Wickham, 2017) suite of packages, including chaining of commands with the pipe-operator commonly associated with the tidyverse workflow. For
spatial representation, R makes use of the *simple features* implementation for R (*sf*; Pebesma (2018)).

Hydrostreamer objects have class *HS*, which are essentially standard R data frames, and which can be modified with any function that works on standard data frames. In a *HS data.frame* object, each row is either a source or target zone which is described by variables and timeseries in *list columns*. Each *HS*-object contains at least a unique ID (riverID for target zones, or zoneID for source zones), and a timeseries column. Depending on the usage, the object can also have a number of other
columns with variables such as topological network information (previous and next river segments) for the routing algorithms, names of monitoring stations, and various timeseries (e.g. runoff, discharge, control, observation timeseries) and variables which are used in the interpolation step. As the *HS* object is a data.frame, additional columns with information the user wants to include can be added. For all the functions which add or modify *HS* specific columns, see Appendix A, Figure A1.

Each timeseries is stored in a *list column*, where each element of the list is a *data.frame* giving the timeseries for the target or source zone in question. Each of these tables are structured so that each row is a timestep, for which the date is given by a column named *Date*. In runoff or discharge timeseries, each additional column is the estimate of an ensemble member. For control and observation timeseries, the table may contain only one column in addition to *Date*.

The river network structure follows a hierarchical node-link network, where each river segment is represented by a node
which has links to previous and next river segments. This data model is simple and intuitive. Demir and Szczepanek (2017) find that this type of river network representation is generally more performant in different types of queries than alternative network representations. The adjacency information is stored in *HS* as list columns NEXT and PREVIOUS, which stores all

the river segment IDs which flow into the segment in question, and which segments are topologically immediately downstream from it.

## 4    Case study method

We conducted a case study in the Sesan, Sekong and Srepok basins (3S from now on) in Southeast Asia to demonstrate hydrostreamer functionality. The 3S's are major transboundary tributaries of the Mekong River, located in Laos, Cambodia and Vietnam. The area is influenced by the Southwest Monsoon, leading to distinct dry and wet seasons. The area is characterized by poor data coverage. We performed downscaling of 15 off-the-shelf global runoff products obtained from the ISIMIP 2a experiment (Gosling et al., 2017), providing an example use case where the scale (downscaling) and zonation
(downscaling to non-conforming target units) effect are addressed with hydrostreamer. We compared the performance of downscaled and routed discharge against the streamflow records in 10 hydrological stations obtained from the Mekong River Commission (MRC, 2017), and against the performance of two free and open global streamflow benchmark datasets. The data record extends from 1985 to 2008, with variable periods at each station.  The following sub-sections detail the data used, performance measurement and three conducted experiments, each building upon the previous.

### 4.1    Data and pre-processing

The experiments build on three main data sources and two distributed global discharge estimates as benchmarks. First, we used the aforementioned ISIMIP 2a data archive (accessed in August 2018). We obtained all total runoff (variable "mrro" in the ISIMIP archive) and discharge ("dis") timeseries, modelled with the variable social forcing ("varsoc") scenario, available in the archive in monthly timestep. When monthly data products were not available, we used the daily product, and
aggregated them to monthly averages. The obtained datasets are summarised in Table 3. We did not use the products forced with WATCH dataset, as it only extends until the end of 2001 and would have meant discarding some of our observation stations with records only after 2001. In total, we obtained from the archive products from 10 global hydrological models and land-surface models (from now on, both referred to as GHM). From the total of 24 runoff products, we only use those which also provided discharge output (n = 15). The ISIMIP outputs are delivered with a spatial resolution of 0.5° degrees
(approximately 55 km at the equator).

**Table 3. ISIMIP 2a total runoff and discharge datasets obtained from the ISIMIP data repository.**

| | Climate forcing | | |
|---|---|---|---|
| | GSWP3 [a] | PGFv2 [b] | WFDEI [c] |

| Model | Runoff | Discharge | Runoff | Discharge | Runoff | Discharge |
|---|---|---|---|---|---|---|
| CARAIB | x | | x | | x | |
| DBH | x | x | x | x | x | x |
| DLEM | x | | x | | x | |
| H08 | x | x | x | x | | x |
| LPJmL | x | x | x | x | x | x |
| MATSIRO | x | x | x | x | x | x |
| PCR-GlobWB | x | x | | x | x | x |
| VEGAS | x | | | | x | |
| VIC | | | x | | | |
| WATERGAP2 | | x | x | x | x | x |

ᵃ Global Soil Wetness Project Phase 3, http://hydro.iis.u-tokyo.ac.jp/GSWP3/
ᵇ Updated version of Sheffield et al. (2006)
ᶜ WATCH Forcing Data – ERA-Interim (Weedon et al., 2014)

The runoff timeseries were downscaled to the HydroSHEDS 30 arc-second resolution river network for Asia (Lehner and Grill, 2013). The total size of the river network within the 3S was 2115 river segments with median length of 5055 meters. To accommodate the evaluation of areal interpolation techniques, we also obtained the HydroSHEDS 30 arc-second resolution flow direction raster from which the river network was derived from. We likewise obtained the HydroSHEDS DEM in order to derive an ancillary variable (see Sect. 4.3).

Similarly to the runoff and discharge GHM timeseries, the daily observed streamflow for the 10 MRC hydrological stations were aggregated to monthly by taking the mean monthly streamflow. For comparison, we additionally use two recent global streamflow products: GRADES (Global Reach-Level A Priori Discharge Estimates for SWOT; Lin et al., 2019) and GLOFAS reanalysis streamflow dataset (GLObal Flood Awareness System; Alfieri et al. 2020). GRADES is provided in zones similar to the HydroSHEDS river network product, albeit derived from a higher resolution DEM. GLOFAS comes as a global grid with 0.1° resolution (~11km at the equator). Both datasets are provided with a daily timestep and were also aggregated to monthly means.

Figure 1A shows the 3S basin and the used HydroSHEDS dataset overlaid on the 0.5° model grid used in the ISIMIP data. The figure additionally shows the locations of the monitoring stations, and the basins they drain from.

## 4.2 Performance measurement

We assessed the performance of the hydrostreamer streamflow predictions and all benchmark datasets to the observation timeseries using commonly used metrics:

1) Root mean square error (RMSE), a commonly used model performance metric. In principle, all model averaging techniques in hydrostreamer minimise RMSE

2) Percent bias (PBIAS), used to estimate model bias in relative terms: mean error standardized to mean observed discharge.

3) Nash-Sutcliffe Efficiency (NSE; Nash and Sutcliffe, 1970), a commonly used performance metric using mean observed streamflow as a benchmark.

4) Coefficient of determination ($R^2$), a standard measure of correlation of dynamics.

5) Kling-Gupta Efficiency (Gupta et al., 2009), a multi-objective metric composed of mean error, variability, and dynamics.

## 4.3 Experiments

We conducted three experiments building upon one another. In the first experiment, we performed downscaling of the total runoff inputs using AWI (for DEM-delineated catchment areas, as well as Voronoi diagram-based delineation), DM (DEM-derived catchments with an ancillary dasymetric variable), and Area-to-Line interpolation (without dasymetric variable). As dasymetric variable, we used a recently developed topographic index DUNE (Dissipation along unit length; Loritz et al., 2019) that is capable of distinguishing different runoff formation regimes and is computed from the HydroSHEDS DEM. We used instantaneous routing for this experiment, because the flow timing error $M$ (Eq. 9) was found to be insignificant considering other potential sources of error (0.028 for the most upstream location, and 0.004 for the 3S basin on average). The most representative river segment was selected from the HydroSHEDS network for each monitoring station location based on comparison to the location on actual river network. For assessment of global model performance, we likewise selected the grid cell which best represents the monitoring station in the low resolution DDM30 river network (Döll and Lehner, 2002) used in ISIMIP framework. We expected that the downscaled Hydrostreamer timeseries should perform at least as well as, or better than, the discharge timeseries from the GHMs due to better representation of the drainage basins associated with each monitoring station.

In the second experiment, we performed model averaging at the monitoring stations and assessed how the uncertainty related to the model averaging weights affects performance of the optimized MMC. In particular, we use the Constrained Least Squares (CLS) technique (Diks and Vrugt, 2010). CLS is constrained to positive weights only, and the sum of weights must equal to one – this means that the MMC timeseries will never have higher or lower discharge than any individual ensemble member. The combinations are performed multiple times with different training periods to assess the uncertainty in the model averaging weights. We used three sampling strategies (each available in hydrostreamer) for the selection of the training period: 1) random selection of 50% of all the timesteps in observation record (performed 100 times for all stations), 2) random selection of 50% of calendar years in the observation record (performed 50 times for all stations), and 3) training combinations for each calendar month separately, with random 50% of available timesteps for each month (performed 50

times for all stations). The timesteps in the observation record not included in the training period were used for model evaluation. We also created 10'000 random combinations with a multi-stage sampling technique, first randomizing the *n* number of models to include, second picking *n* random models, and third randomising positive weights among the randomized model selection using uniform distribution with weights summing to unity in order to have comparable

constraint in the randomisation as we have in CLS.

Finally, in the third experiment, we regionalize the optimized MMC weights at each monitoring station and evaluate their performance at the other monitoring stations the same river.

## 5    Case study results and discussion

The results of each of the three experiments are explored and discussed in the following subsections.

### 5.1    Experiment 1: downscaling (interpolation, routing)

The four tested downscaling methods (see Sect. 4.3) show negligible difference across all used performance metrics, when averaged over monitoring stations (see Appendix A, Table A1). Station-wise, there are very small differences in all stations except Sesan Upstream-East, where the largest differences in performance between downscaling methods are up to 0.40 for NSE, 0.10 for KGE, both when downscaling H08 forced with PGFv2. Across the entire 3S Basin, the difference between

downscaling methods becomes smaller as the basin size increases. This is expected: with the chosen instantaneous routing method the only difference in discharge comes from the basin boundary, since all runoff from the middle of the basin instantly flows through the station. As the basin size increases, the proportion of runoff contribution from the catchments at the basin boundary becomes increasingly small, and thus is shown in decreasing difference in performance metrics. This is in line with Cunha et al. (2012), who find that in ensemble modelling uncertainty gets smaller with increasing basin size.

Because the differences between the downscaling are so small apart from the Sesan Upstream-East station, we opt to continue the analysis on the simplest downscaling method: Area-to-Line interpolation. It should be noted, however, that Virkki (2019) showed that Area-to-Line interpolation causes larger uncertainties in the reach-level than approaches using reach-specific catchment areas. Furthermore, Kallio et al. (2019) found in a study that included 126 catchments with natural flow regime that DM using DUNE as ancillary variable does improve the performance of downscaling in topographically

varying terrain.

The performance of individual downscaled GHMs varies much more than the performance between tested downscaling methods (see Appendix A, Tables A2-A4). Compared to the discharge output from GHMs, the downscaled ones are similar or better in their performance, as seen in Figure 3. Moriasi et al. (2015) recommend that for watershed scale models at monthly temporal resolution acceptable model performance is $R^2 > 0.60$, NSE > 0.50, and PBIAS ≤ ±15 %. Using this

criteria, we find that they are fulfilled in 47 % and 55 % ($R^2$), 22 % and 43 % (NSE), and 27 % and 53 % (PBIAS) of cases

in downscaled GHMs and GHMs, respectively (see Table 4). Volume-wise the downscaled estimates fare better with mean (across all 15 GHM-climate forcing pairs, and all monitoring stations) PBIAS of -0.3% against GHM's 11.2%. The difference and direction in performance is visualized in Figure 4, confirming our expectation that downscaling does improve the performance of GHM outputs.

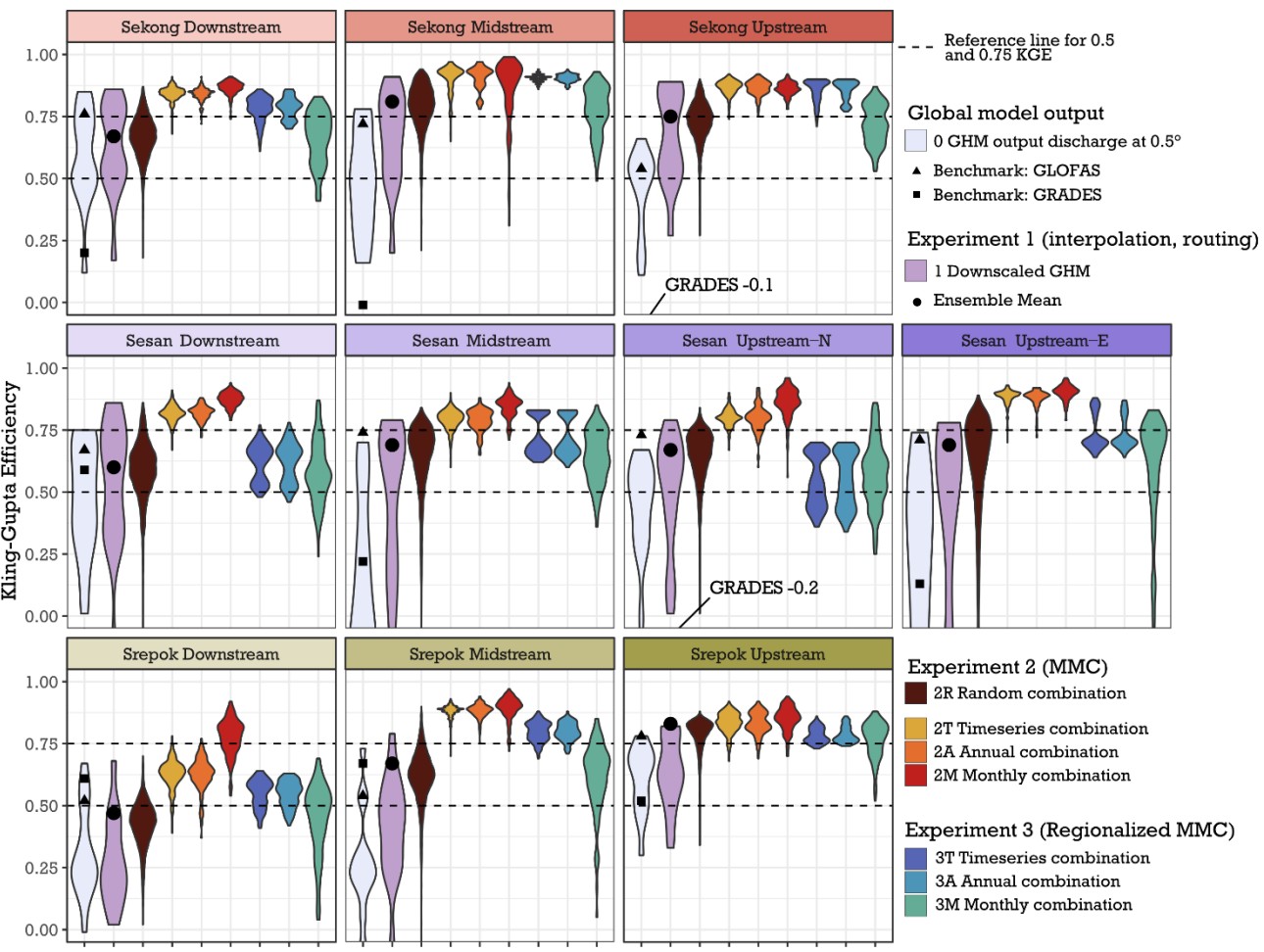

Figure 3. Comparison of Kling-Gupta Efficiency compared to observed streamflow from the MRC for global model output and three experiments. Global model output: discharge from ISIMIP GHMs at 0.5° resolution (n=15), with results from benchmark products; Experiment 1: downscaled ISIMIP runoff; Experiment 2: Multi-Model Combination (MMC) derived from random combinations, and the three sampling strategies for CLS (i.e. Constrained Least Squares, see Table 1) model averaging (2T, 2A, 2M); and Experiment 3: using regionalized MMC weights derived in Experiment 2 at the gauges in the same river (Sekong, Sesan, or Srepok). KGE is shown for testing period for Experiments 2 and 3, and for the entire timeseries for GHM discharge and Experiment 1.

**Table 4. Proportion of test cases (separate predictions at each station) satisfying performance criteria from Moriasi et al. (2015) for discharge and downscaled runoff from GHMs, and for the different combination strategies tested. KGE is added to the criteria as an alternative to NSE, and to allow comparison to Figure 3. Values are shown for test period for 2T, 2A and 2M, and for the entire timeseries for everything else. Proportions > 50% shown in bold. BM stands for benchmark dataset.**

| | Type of ensemble | n | Individual ensemble members R2 > 0.6 | NSE > 0.5 | PBIAS ≤ ±15 % | KGE > 0.5 | n | Ensemble mean R2 > 0.6 | NSE > 0.5 | PBIAS ≤ ±15 % | KGE > 0.5 |
|---|---|---|---|---|---|---|---|---|---|---|---|
| BM | GLOFAS | 10 | **90%** | **90%** | 30% | **100%** | - | - | - | - | - |
| BM | GRADES | 10 | 40% | 10% | 10% | 40% | - | - | - | - | - |
| 0 | Global model | 150 | 47% | 22% | 27% | 43% | 10 | **70%** | **60%** | 30% | **80%** |
| 1 | Downscaled global model | 150 | **55%** | 43% | **53%** | **61%** | 10 | **80%** | **80%** | 50% | **90%** |
| 2R | Random | 100 000 | **72%** | **65%** | **56%** | **83%** | 10 | **80%** | **80%** | 50% | **90%** |
| 2T | Timeseries | 1000 | **99%** | **98%** | **84%** | **99%** | 10 | **100%** | **100%** | **90%** | **100%** |
| 2A | Annual | 500 | **99%** | **97%** | **82%** | **99%** | 10 | **100%** | **100%** | **90%** | **100%** |
| 2M | Monthly | 500 | **98%** | **98%** | **91%** | **99%** | 10 | **90%** | **90%** | 50% | **90%** |
| 3T | Regionalized timeseries | 2400 | **90%** | **89%** | **58%** | **92%** | 10 | **100%** | **100%** | **60%** | **100%** |
| 3A | Regionalized annual | 1200 | **91%** | **90%** | **59%** | **92%** | 10 | **100%** | **100%** | **70%** | **100%** |
| 3M | Regionalized monthly | 1200 | **67%** | **60%** | **51%** | **82%** | 10 | **90%** | **90%** | 50% | **80%** |

Comparing to the openly available benchmark products GRADES and GLOFAS, the downscaled GHMs fare reasonably well. While the individual ensemble members have large spread in their performance, being often worse than either of the benchmarks, the ensemble mean provides consistent good performance with KGE > 0.5 in all stations except Srepok Downstream (Figure 3). The ensemble mean is considerably better performing than GRADES, in all stations but Srepok

10  Downstream and Midstream. GLOFAS performs better than the ensemble mean in 6 of the 10 stations.

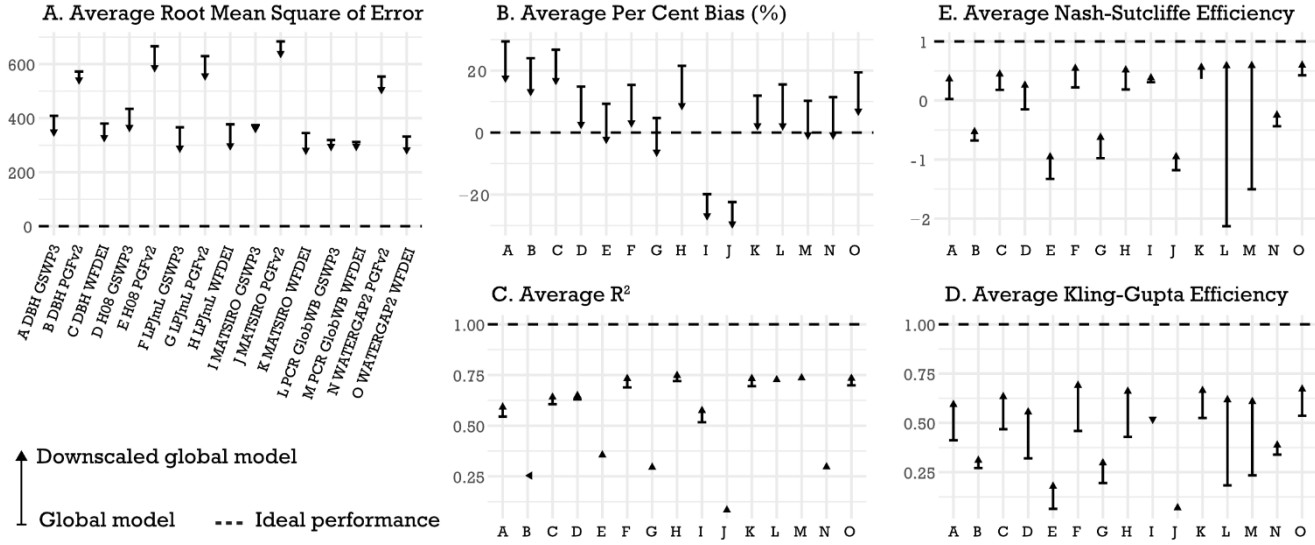

**Figure 4. Comparison of the performance of GHMs and downscaled GHMs averaged over all 10 monitoring stations.**

## 5.2    Experiment 2: Multi-Model Combinations at point locations

In the second experiment we performed model averaging on the 15-member ensemble with three combination strategies. For all stations, the timeseries and annual combination strategies produce very similar distribution in performance (Figure 3, distributions 2T and 2A). Monthly combination (Figure 3, distribution 2M) strategy can, however, produce better performance at point locations for PBIAS, which is lower than the threshold (PBIAS < 15%) in 91% of all combinations (Table 4). However, when taking an ensemble mean from the 50 monthly combinations for each station, PBIAS threshold is satisfied in only half of the stations – and on the other hand the ensemble mean from timeseries or annual combinations perform considerably better. Comparing against the benchmarks, GRADES and GLOFAS, we see that all of the combination strategies can produce better performance (Figure 3), with only a small minority of optimised MMC combinations showing worse performance for KGE.

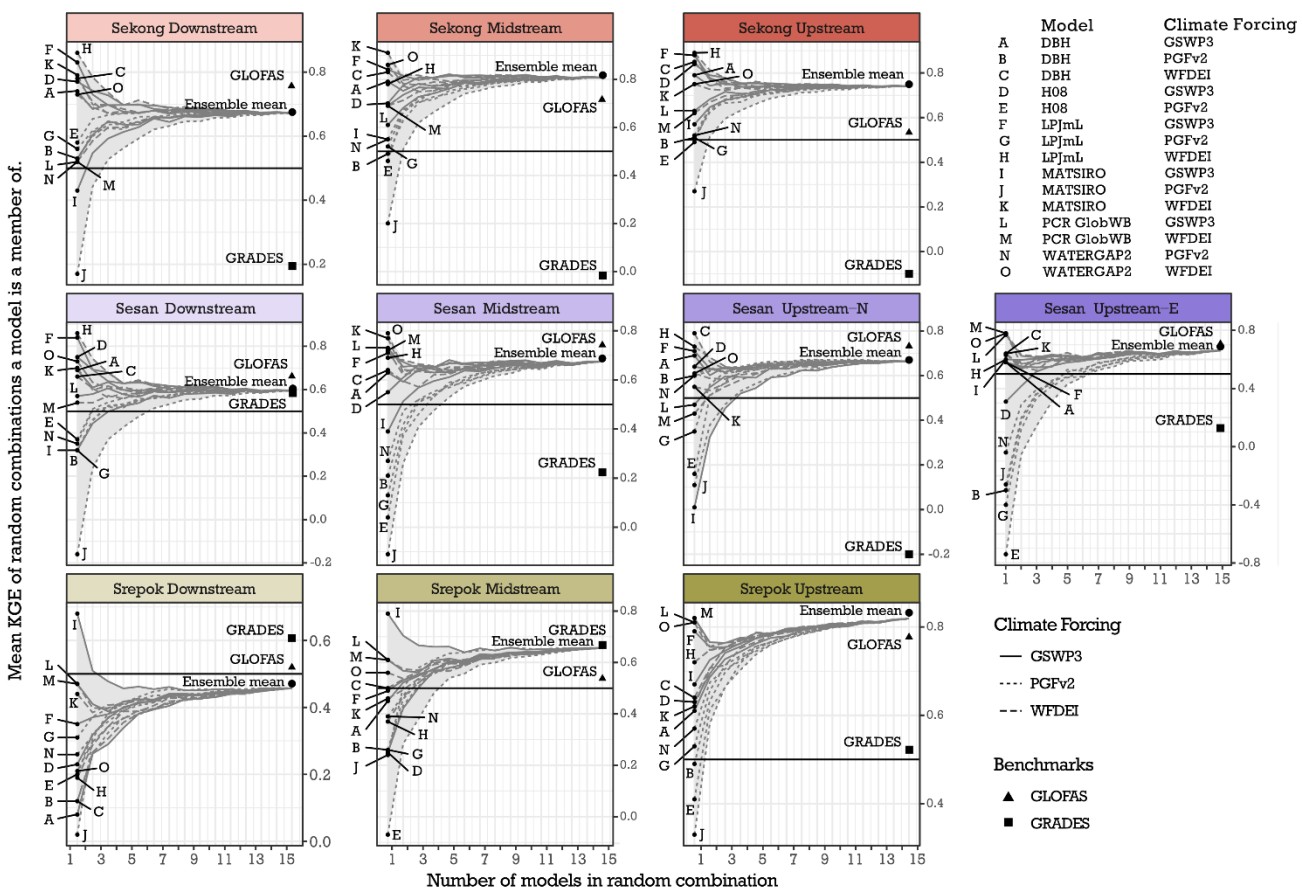

**Figure 5. KGE performance of individual downscaled runoff ensemble members, and the mean KGE of the random ensembles each model is a member of. The two benchmarks GRADES, GLOFAS, and the ensemble mean are marked for comparison.**

5     When looking at individual ensemble members in Figure 5, and their performance with random MMC weights, we see that the models have highly variable performance at different stations. The performance of individual ensemble members varies between stations (Figure 5), and between indicators (Appendix A Table A2). MATSIRO in particular shows high sensitivity to climate forcing; the performance when forced with PGFv2 has the largest mean RMSE, but forced with GSWP3 results in the smallest RMSE. In general, models forced with PGFv2 perform considerably worse at the 3S than when forced with

10     WFDEI or GSWP3 (see Appendix A Table A3).

The ensemble mean is robust throughout the basin; it is among the best performing individual members for most of the stations, and for Srepok Upstream it performs better than any single ensemble member. We can further infer from Figure 5 that the skill of the ensemble mean stabilizes at around 10-12 random ensemble members.

**5.3     Experiment 3: Regionalisation of Multi-Model Combinations – prediction in ungauged basins**

In the third experiment we tested the applicability of regionalizing weights derived at one station to the other stations in the basin. We used weights derived from the stations with direct upstream or downstream linkage – Sekong, Sesan, and Srepok stations separately (refer to Figure 1A). Our results suggest that the performance of regionalized model averaging weights is
variable, as seen in Figure 3. Regionalized timeseries and annual model averaging strategies produce commonly higher performance than the ensemble mean or the distribution of the random ensemble combinations, and in some stations can produce similar performance to the optimum for that station. This is desirable, as regionalization of MMC weights would make no sense if simple ensemble mean would perform better. We explored the distribution of weights at different stations and found that Sekong and Srepok stations produce entirely different weighting of ensemble members. Sesan stations are
somewhere between, with Sesan Downstream showing similar MMC weights to Sekong stations, and Sesan Upstream-East similar to Srepok rivers (the distribution of weights not shown). Sesan Midstream and Upstream-North appear unique in their sets of MMC weights. This is clearly seen in the distribution of performance of the Sesan stations in Figure 3; there are clear clusters of performance from weights from the other stations, whereas Sekong and Srepok stations give a more uniform distribution in regionalized MMC performance.

The distribution of performance of the regionalized weights from monthly combinations is a clear case of overfitting – the distribution of performance (3M – the rightmost distribution in each facet in Figure 3) is very large and is similar or worse than the ensemble mean (Table 4). This is natural, since in the monthly weighting strategy we develop a set of 12 weights (one set for each month of the year) instead of a single set with timeseries and annual combination strategies. This suggests that monthly combinations are more useful for point optimisations, but are not advantageous for use cases requiring
regionalisation.

**6     Planned future developments**

Hydrostreamer has been developed with two goals in mind: first, to support non-expert audiences in access to hydrological model data for their specific use cases, and secondly to improve the usability of existing off-the-shelf hydrological products.
Hydrostreamer can help non-experts in deriving hydrological variables they need by providing the means to avoid the pitfalls of hydrological modelling and to use data products prepared by experts. Hydrostreamer enables this by providing one way of dealing with MAUP (Goodchild and Lam, 1980) – the hydrological data product can be transformed to fit the analysis at hand. Using data products prepared and validated by experts can help in building confidence in the analysis results. With reference to the evaluation framework for environmental modelling developed by Hamilton et al. (2019), avoiding rushed
modelling by inexperienced modellers can improve the confidence in several project-level elements of that framework – 1) efficiency by reducing time needed to produce estimates, 2) credibility by using outputs from professional hydrologists, 3)

legitimacy by reducing bias when using multiple input runoff estimates, and 4) accessibility when using freely available runoff products.

The case study showed that overall, using global hydrological data products can produce results comparable or better than openly available streamflow products with a global scope, *even when the simplest possible case* – Area-to-Line Interpolation – is used. We attribute this to a better representation of the drainage network than in the 0.5° GHMs. Version 1.0 of hydrostreamer has limitations, however, some of which we mention here. The biggest limitation of the current implementation is that DM and PP currently only allow temporally static weighting, similar to AWI. There are, however, many potential ancillary variables which may guide DM and PP which could be input with a timeseries. We plan that future versions of hydrostreamer will support timeseries for the dasymetric and pycnophylactic variables allowing dynamic interpolation.

The implemented instantaneous and constant flow velocity river routing methods are simpler than the commonly used methods (e.g. RAPID and MizuROUTE implement Muskingum and kinematic wave routing) but similar to a number of global hydrological models (Telteu et al., 2021). These two options are attractive due to their simplicity; the instantaneous routing solution does not have any parameters, and the constant velocity has only one (flow velocity). The Muskingum-Cunge routing option may be more attractive for advanced users and when the simpler alternatives are not reasonable, but comes at a cost of estimating Manning's roughness coefficient, bed slope and river width. These may be estimated by the physical properties of the river segments, using e.g. a DEM, provided that such data is available. The routing solutions in hydrostreamer do not, currently, include a reservoir or a lake model, which limits their applicability. Our case study area is devoid of large-scale dams during the simulation period, apart from Houay Ho and Yali built in 1999 and 2002, respectively. Models tend to be skilful in compensating for hydropower even when they are not represented in the models (as we can see from the high performance of MMC combinations in Sesan Midstream and Downstream stations located downstream from Yali). This, however, leads to overfitting and not a true representation of the parameters (Dang et al., 2020). In hydrostreamer the relevant parameters are the MMC weights (if MA is applied) and any parameters required by the routing model. In our experimental results this is limited since Yali has been operational for only the last 6 years (out of a total of 23) of the streamflow record, and influences only Sesan Midstream and Downstream stations. Houay Ho is located on a small tributary of Sekong and does not have a large influence on the flow regime. We plan to add simple reservoir and lake models into the routing methods, however, in hydrostreamer v1.0 reservoirs can be represented through setting a boundary condition for the river segment in which a reservoir outlet is located.

The limitations in the model averaging step are most substantial in the regionalization component. Hydrostreamer currently only supports regionalizing weights to the upstream segments from a monitoring station up to the next monitoring station, defaulting to ensemble mean on every segment without a downstream dam. We plan to address this by adding further regionalization options, for instance, based on proximity and similarity of river segments.

## 7    Conclusion

In this paper we presented *hydrostreamer* v1.0, an R package designed to improve usability of hydrological data products and to support the use of hydrological data products by non-experts. Hydrostreamer does this particularly by addressing the Modifiable Area Unit Problem – pre-existing data products often arrive at a spatial aggregation or incompatible enumeration units which are not optimal for user's analysis case. This article includes an overview of the concepts and workflow hydrostreamer is built upon: (areal) interpolation, routing, and model averaging. There are several features in hydrostreamer which are not available in other software solutions in R: advanced areal interpolation methods dasymetric mapping (for both area-to-area and Area-to-Line interpolation), pycnophylactic interpolation for polygon networks, and a combined pycnophylactic-dasymetric interpolation specifically designed for hydrological variables. Further, there are no other vector-based river routing solutions available for R. Hydrostreamer also facilitates data assimilation via model averaging when observation data is available.

To test the capabilities of hydrostreamer, we performed a case study downscaling an ensemble of global runoff products onto HydroSHEDS 15 arc-second river network. We show that an ensemble of coarse resolution global hydrological products can be used to produce locally accurate streamflow timeseries – even with the simplest forms of areal and Area-to-Line Interpolation. This we attribute to addressing MAUP by better representation of the drainage network and catchment areas. We find that model averaging weights can be transferred to ungauged locations, but with some limitations such as non-negativity of the weights, and sufficient similarity of catchments. This represents a clear future research topic. We further find that an ensemble mean of global hydrological models can produce an adequate estimate for streamflow, at least for monthly timestep.

Hydrostreamer fills a niche where streamflow data is needed quickly, but limited resources (skill, time, money, input data) are available to set up a new modelling exercise. Using hydrostreamer, reasonable quality streamflow estimates can be extracted from existing runoff products with the addition of only a river network and historical streamflow records for model averaging. Hydrostreamer v1.0 is open source and available under MIT licence from GitHub: http://github.com/mkkallio/hydrostreamer/.

# Appendix A

**Table A1. The mean goodness-of-fit measurements of the tested downscaling methods at each monitoring station, averaged across all GHM-climate forcing pairs. Ordered by KGE.**

| Method | Station | RMSE | PBIAS % | NSE | $R^2$ | KGE |
|---|---|---|---|---|---|---|
| AWI (Thiessen Polygons) | Sekong Downstream | 911 | -18.2 | 0.48 | 0.60 | 0.62 |
| Area-to-Line | Sekong Downstream | 911 | -18.4 | 0.48 | 0.60 | 0.62 |
| DM | Sekong Downstream | 911 | -18.5 | 0.48 | 0.60 | 0.62 |
| AWI (DEM delineated) | Sekong Downstream | 911 | -18.5 | 0.48 | 0.60 | 0.62 |
| AWI (DEM delineated) | Sekong Midstream | 468 | -1.3 | 0.47 | 0.63 | 0.66 |
| AWI (Thiessen Polygons) | Sekong Midstream | 468 | -1.3 | 0.47 | 0.63 | 0.66 |
| DM | Sekong Midstream | 468 | -1.4 | 0.47 | 0.63 | 0.66 |
| Area-to-Line | Sekong Midstream | 470 | -0.5 | 0.47 | 0.63 | 0.66 |
| DM | Sekong Upstream | 328 | -4.4 | 0.43 | 0.58 | 0.67 |
| AWI (DEM delineated) | Sekong Upstream | 325 | -5.9 | 0.44 | 0.59 | 0.67 |
| AWI (Thiessen Polygons) | Sekong Upstream | 325 | -6.1 | 0.44 | 0.59 | 0.67 |
| Area-to-Line | Sekong Upstream | 323 | -8.2 | 0.45 | 0.59 | 0.67 |
| DM | Sesan Downstream | 555 | -17.2 | 0.29 | 0.51 | 0.53 |
| Area-to-Line | Sesan Downstream | 555 | -18.1 | 0.28 | 0.51 | 0.52 |
| AWI (DEM delineated) | Sesan Downstream | 556 | -18.1 | 0.28 | 0.51 | 0.52 |
| AWI (Thiessen Polygons) | Sesan Downstream | 556 | -18.1 | 0.28 | 0.51 | 0.52 |
| AWI (DEM delineated) | Sesan Midstream | 386 | 2.0 | -0.09 | 0.42 | 0.48 |
| Area-to-Line | Sesan Midstream | 385 | 2.2 | -0.08 | 0.43 | 0.48 |
| AWI (Thiessen Polygons) | Sesan Midstream | 386 | 2.4 | -0.09 | 0.42 | 0.48 |
| DM | Sesan Midstream | 387 | 2.7 | -0.09 | 0.42 | 0.47 |
| DM | Sesan Upstream-N | 84 | -21.6 | 0.30 | 0.62 | 0.52 |
| AWI (Thiessen Polygons) | Sesan Upstream-N | 85 | -22.5 | 0.30 | 0.62 | 0.51 |
| AWI (DEM delineated) | Sesan Upstream-N | 85 | -23.2 | 0.31 | 0.62 | 0.51 |
| Area-to-Line | Sesan Upstream-N | 86 | -24.3 | 0.30 | 0.62 | 0.50 |
| DM | Sesan Upstream-E | 78 | -3.8 | -0.49 | 0.54 | 0.34 |
| AWI (DEM delineated) | Sesan Upstream-E | 78 | -3.4 | -0.49 | 0.54 | 0.34 |
| AWI (Thiessen Polygons) | Sesan Upstream-E | 79 | -2.0 | -0.53 | 0.54 | 0.32 |
| Area-to-Line | Sesan Upstream-E | 80 | -0.9 | -0.59 | 0.54 | 0.30 |
| DM | Srepok Downstream | 764 | 47.5 | -0.11 | 0.49 | 0.28 |
| AWI (DEM delineated) | Srepok Downstream | 767 | 48.1 | -0.12 | 0.49 | 0.28 |
| Area-to-Line | Srepok Downstream | 767 | 48.2 | -0.12 | 0.49 | 0.28 |
| AWI (Thiessen Polygons) | Srepok Downstream | 767 | 48.2 | -0.12 | 0.49 | 0.28 |
| AWI (DEM delineated) | Srepok Midstream | 237 | 15.9 | 0.01 | 0.59 | 0.45 |

| | | | | | | |
|---|---|---|---|---|---|---|
| DM | Srepok Midstream | 238 | 16.2 | 0.00 | 0.59 | 0.45 |
| AWI (Thiessen Polygons) | Srepok Midstream | 240 | 16.8 | -0.02 | 0.59 | 0.44 |
| Area-to-Line | Srepok Midstream | 246 | 18.8 | -0.07 | 0.59 | 0.41 |
| AWI (Thiessen Polygons) | Srepok Upstream | 73 | -5.9 | 0.35 | 0.55 | 0.64 |
| AWI (DEM delineated) | Srepok Upstream | 73 | -6.2 | 0.35 | 0.54 | 0.64 |
| DM | Srepok Upstream | 75 | -3.2 | 0.33 | 0.54 | 0.64 |
| Area-to-Line | Srepok Upstream | 76 | -1.5 | 0.31 | 0.55 | 0.63 |

**Table A2. The mean goodness-of-fit measures of Area-to-Line downscaling method for all included GHM-climate forcing pairs. The values are averaged over all 10 monitoring stations.**

| Model | Climate Forcing | RMSE | PBIAS % | NSE | R2 | KGE |
|---|---|---|---|---|---|---|
| LPJmL | GSWP3 | 280 | 2.4 | 0.58 | 0.74 | 0.70 |
| WATERGAP2 | WFDEI | 274 | 6.2 | 0.64 | 0.75 | 0.68 |
| MATSIRO | WFDEI | 273 | 1.5 | 0.60 | 0.74 | 0.68 |
| LPJmL | WFDEI | 289 | 8.0 | 0.56 | 0.76 | 0.67 |
| DBH | WFDEI | 320 | 16.1 | 0.49 | 0.65 | 0.65 |
| PCR-GlobWB | GSWP3 | 287 | 1.4 | 0.63 | 0.74 | 0.63 |
| PCR-GlobWB | WFDEI | 287 | -1.5 | 0.63 | 0.75 | 0.62 |
| DBH | GSWP3 | 340 | 16.8 | 0.40 | 0.60 | 0.61 |
| H08 | GSWP3 | 357 | 1.9 | 0.30 | 0.66 | 0.57 |
| MATSIRO | GSWP3 | 353 | -27.6 | 0.42 | 0.59 | 0.51 |
| WATERGAP2 | PGFv2 | 497 | -1.5 | -0.21 | 0.31 | 0.40 |
| DBH | PGFv2 | 533 | 12.5 | -0.49 | 0.25 | 0.32 |
| LPJmL | PGFv2 | 548 | -7.1 | -0.59 | 0.31 | 0.31 |
| H08 | PGFv2 | 579 | -2.9 | -0.92 | 0.37 | 0.19 |
| MATSIRO | PGFv2 | 631 | -30.2 | -0.91 | 0.10 | 0.08 |

**Table A3. The mean performance of climate forcing datasets, averaged over all 10 monitoring stations and all GHMs.**

| Climate Forcing | RMSE | PBIAS % | NSE | R2 | KGE |
|---|---|---|---|---|---|
| WFDEI | 328 | 4.9 | 0.20 | 0.62 | 0.49 |
| GSWP3 | 351 | -1.5 | 0.14 | 0.57 | 0.47 |
| PGFv2 | 537 | -5.9 | -0.40 | 0.29 | 0.32 |

**Table A4. The mean performance of GHMs, averaged over all climate forcing datasets and all 10 monitoring stations. It should be noted that PCR-GlobWB does not include a version forced with PGFv2, which in this basin has the highest error.**

| Model | RMSE | PBIAS % | NSE | R2 | KGE |
|---|---|---|---|---|---|
| PCR-GlobWB* | 287 | 0.0 | 0.63 | 0.74 | 0.62 |
| LPJmL | 373 | 1.1 | 0.18 | 0.60 | 0.56 |
| WATERGAP2 | 386 | 2.3 | 0.21 | 0.53 | 0.54 |
| DBH | 398 | 15.1 | 0.13 | 0.50 | 0.52 |
| MATSIRO | 419 | -18.8 | 0.04 | 0.48 | 0.42 |
| H08 | 468 | -0.5 | -0.31 | 0.51 | 0.38 |

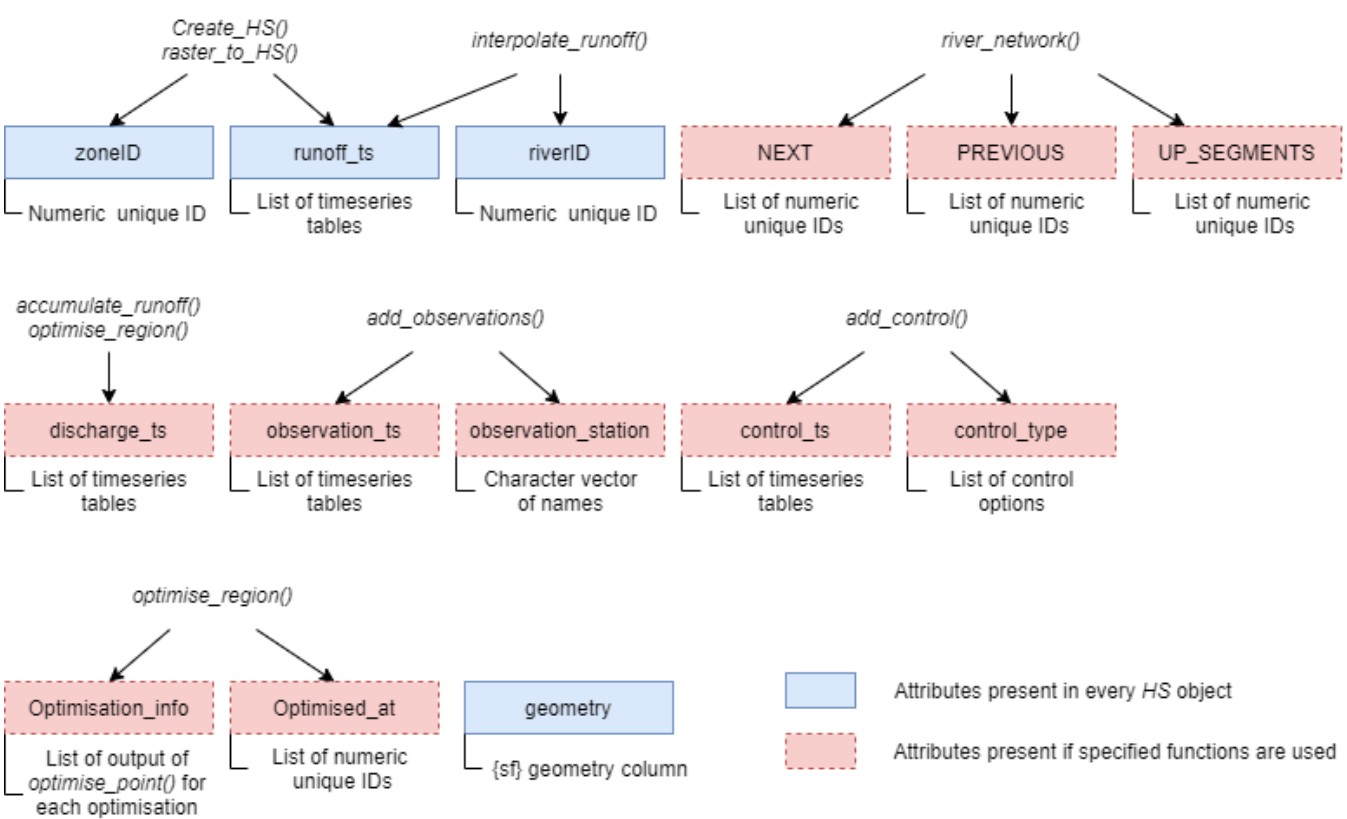

**Figure A1. Attribute columns which may be added by hydrostreamer functions to an *HS* object, and the functions which include them.**

# Appendix B

**Conceptual illustration of the constant flow velocity routing method implemented in hydrostreamer v1.0.**

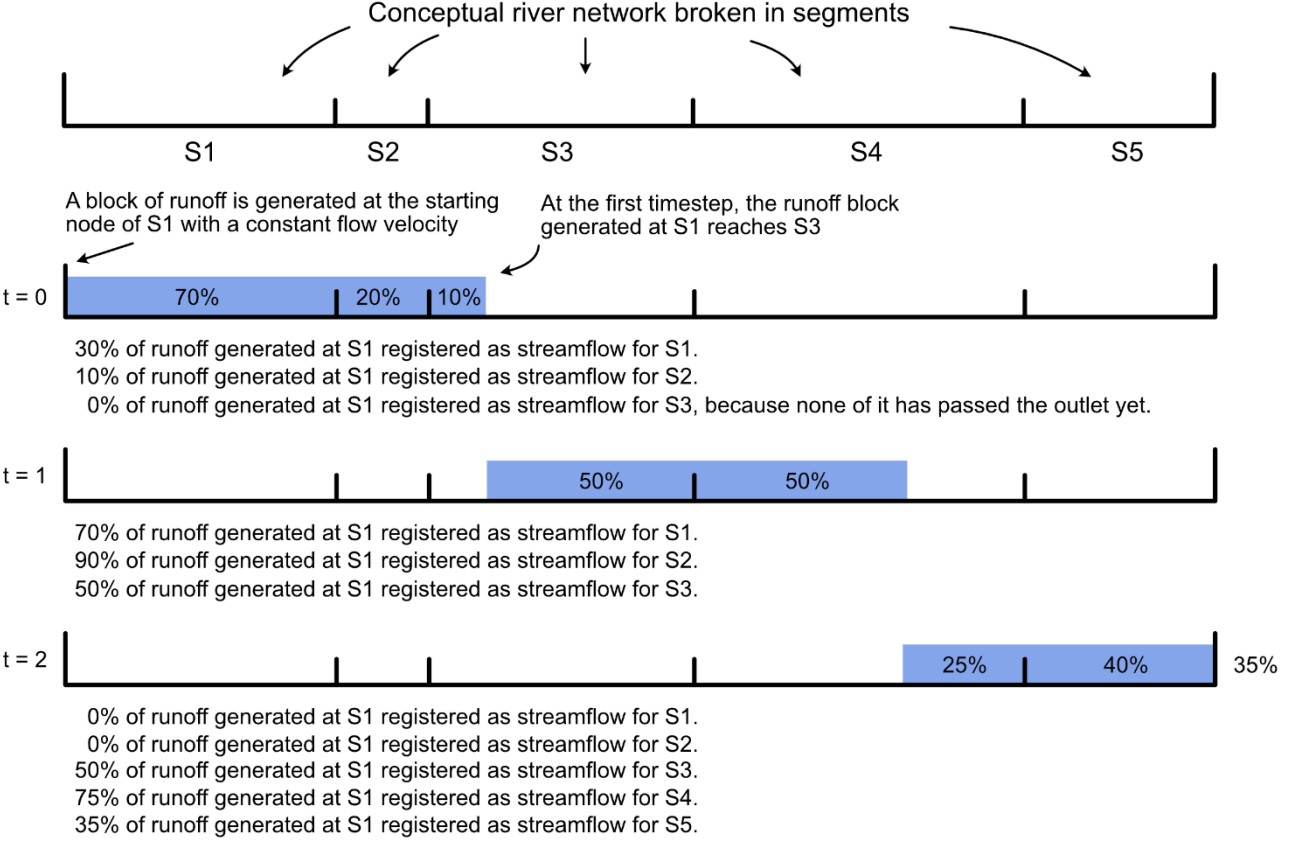

**Figure B1. Conceptual representation of the constant velocity algorithm, showing runoff produced at S1 at timestep t = 0, and how it is registered at downstream river segments.**

## Comparison of the three flow routing methods

The three streamflow routing methods were compared for our case study area using LPJmL model forced with GSWP3

10    climate forcing. We ran the constant velocity routing with the default 1 m s$^{-1}$ flow velocity. Muskingum-Cunge was run

using constant Manning's roughness coefficient of 0.03, and with constant slope of 0.00025 for all river segments. For river

width modelling, we used the power-law relationships between discharge and river width from Moody and Troutman (2002).

Performance metrics used in the case study are shown in Table B1. The predicted timeseries are shown in Figure B2. With

the parameters given above, constant velocity routing performs the best. However, at a monthly time scale there is little practical difference between the methods in the study area. The performance of Muskingum-Cunge and constant velocity routing is expected to improve with optimised routing parameters and velocity, respectfully.

**Table B1. Performance metrics of the three routing methods implemented in hydrostreamer for the GHM LPJmL forced with GSWP3 climate dataset. MC stands for Muskingum-Cunge algorithm, Const. for constant velocity routing and Inst. for instantaneous routing.**

| Station | NRMSE % | | | PBIAS % | | | NSE | | | KGE | | | R2 | | |
|---|---|---|---|---|---|---|---|---|---|---|---|---|---|---|---|
| | MC | Const. | Inst. | MC | Const. | Inst. | MC | Const. | Inst. | MC | Const. | Inst. | MC | Const. | Inst. |
| Sekong Downstream | 72.2 | 68.2 | 74.7 | -12 | -12 | -12 | 0.48 | 0.53 | 0.44 | 0.69 | 0.7 | 0.68 | 0.53 | 0.56 | 0.5 |
| Sekong Midstream | 75 | 72.9 | 75.8 | 10.6 | 10.6 | 10.5 | 0.43 | 0.46 | 0.42 | 0.71 | 0.72 | 0.7 | 0.57 | 0.59 | 0.57 |
| Sekong Upstream | 90.8 | 89.3 | 91.6 | -1.8 | -1.8 | -1.8 | 0.17 | 0.2 | 0.16 | 0.43 | 0.43 | 0.42 | 0.24 | 0.25 | 0.23 |
| Sesan Downstream | 76.9 | 74.4 | 78.4 | -10.1 | -10.1 | -10.1 | 0.41 | 0.44 | 0.38 | 0.66 | 0.67 | 0.65 | 0.47 | 0.49 | 0.46 |
| Sesan Midstream | 97.1 | 94.8 | 98.3 | 11.7 | 11.7 | 11.7 | 0.05 | 0.1 | 0.03 | 0.55 | 0.57 | 0.54 | 0.34 | 0.35 | 0.33 |
| Sesan Upstream-E | 126.7 | 124.9 | 126.9 | 2.9 | 2.9 | 2.9 | -0.61 | -0.57 | -0.62 | 0.34 | 0.35 | 0.33 | 0.15 | 0.16 | 0.15 |
| Sesan Upstream-N | 92.9 | 91.7 | 93.1 | -28.6 | -28.7 | -28.6 | 0.13 | 0.15 | 0.12 | 0.46 | 0.46 | 0.46 | 0.31 | 0.33 | 0.31 |
| Srepok Downstream | 105.1 | 110.7 | 114.9 | 62.1 | 64.8 | 64.8 | -0.11 | -0.23 | -0.33 | 0.25 | 0.21 | 0.19 | 0.35 | 0.36 | 0.33 |
| Srepok Midstream | 128.9 | 126 | 131 | 23.7 | 23.7 | 23.7 | -0.67 | -0.59 | -0.72 | 0.31 | 0.33 | 0.29 | 0.22 | 0.24 | 0.21 |
| Srepok Upstream | 106.3 | 104.7 | 106.6 | 4.7 | 4.7 | 4.7 | -0.13 | -0.1 | -0.14 | 0.49 | 0.5 | 0.49 | 0.26 | 0.27 | 0.26 |

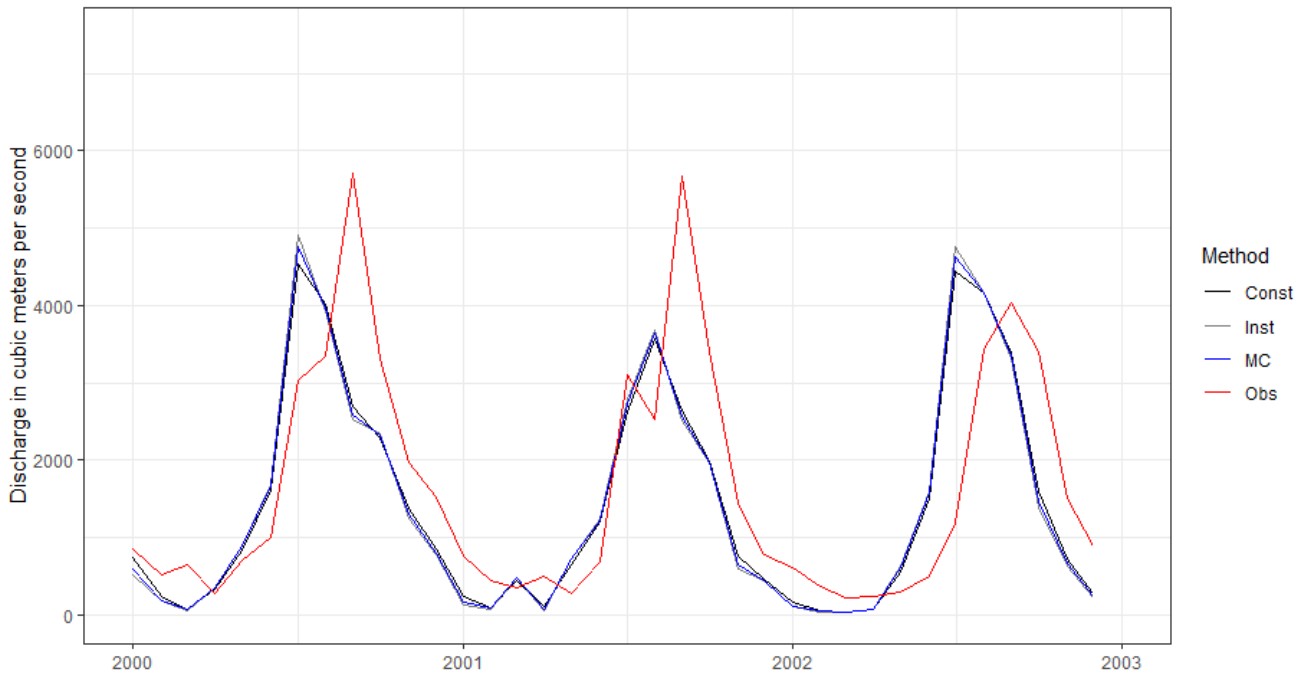

**Figure B2. Predicted monthly LPJmL-GSWP3 flow of the three routing methods at Sekong Downstream Station. Figure showing a sample of years 2000-2002. MC stands for Muskingum-Cunge algorithm, Const. for constant velocity routing,  Inst. for instantaneous routing, and Obs. for observations.**

*Code and data availability*

The ISIMIP data used in this study are available from https://esg.pik-potsdam.de/search/isimip/. Benchmark datasets GRADES (Lin et al., 2019) is available for research purposes at http://hydrology.princeton.edu/data/mpan/MERIT_Basins/ and http://hydrology.princeton.edu/data/mpan/GRADES/,  and GLOFAS (Alfieri et al., 2020) is available from

https://cds.climate.copernicus.eu/cdsapp#!/dataset/cems-glofas-historical. HydroSHEDS data is available from https://hydrosheds.org/. Streamflow data is available from the Mekong River Commission Data Portal at https://portal.mrcmekong.org/. Hydrostreamer v1.0.1 source code for this publication is deposited at https://zenodo.org/record/4739223 with the latest version of the code hosted at github: https://github.com/mkkallio/hydrostreamer. Code and data (except for observed timeseries, due to license constraints) to

reproduce the analysis and the output of downscaling and routing are archived at https://zenodo.org/record/4739212.

*Author contributions*

MKa planned and envisioned the software. MKa wrote the software with input from VV. MKa and JG designed the experiments. MKa wrote the manuscript with input from all the authors.

*Acknowledgements*

We would like to thank the ISIMIP team and all participating modelling teams for making the outputs freely available.

*Financial support*

MKa and VV were funded by the Aalto University School of Engineering Doctoral Programme. MKa additionally received funding from Maa- ja Vesitekniikan Tuki ry. MKu received funding from Academy of Finland funded project WATVUL (grant no. 317320), Emil Aaltonen Foundation funded project 'eat-less-water', and European Research Council (ERC) under the European Union's Horizon 2020 research and innovation programme (grant agreement No. 819202) from which VV received additional funding. JG received funding from an Australian Research Council Discovery Early Career Researcher

Award (project no. DE190100317).

*Competing interests*

The authors declare that they have no conflict of interest.

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
