# Peer review of "Hydrostreamer v1.0 - improved streamflow predictions for local applications from an ensemble of downscaled global runoff products"

_Geoscientific Model Development, 2020_

## Referee Comment (RC1) · Anonymous Referee #1 · 20 Jan 2021

Comments to the Author In this manuscript, a useful tool is developed for non-hydrologist to use runoff products estimated by various land surface models. The tool mainly has three functions: 1) mapping the runoff components from the land surface model to units in the hydrological model, 2) modeling the river routing processes, and 3) assimilation via modeling averaging. The article is well written and well organized. I have a few suggestions which might help make the paper stronger. I think it could be done as a major revision.

1. In the manuscript two mapping methods are provided by the developed tool. The area-to-line interpolation is not looking reasonable to me. In this approach, the intersecting portion of the river line within the source zone is used as average weight. The approach did not respect the actual drainage area controlled by the river line, actually, somehow recalculate the drainage area of each river line during the mapping process. In an extreme case, if a river line flows along the boundary of the grid, then no runoff will contribute to that river line.

2. Two simple methods are offers for river routing. I would suggest adding more routing options for example diffusive wave, hydrological routing approaches. It would be interesting to also assimilate the simulated streamflow from different routing methods, not only using different runoff inputs.

---

## Referee Comment (RC2) · Anonymous Referee #2 · 22 Feb 2021

The study presents an R package, a software library hydrostreamer v1.0 which aims to improve the usability of existing runoff products by addressing the Modifiable Area Unit Problem, and allows nonexperts with little knowledge of hydrology-specific modelling issues and methods to use them for their analyses. The topic is well suited for publication in GMD, however, the manuscript has some unclear reasoning that requires significant revision before the manuscript can be accepted for publication. My major comments are provided as follows.

1. The work was motivated by providing a tool that can be used by non-hydrologist to downscale global runoff products to river-basin scale for follow us analysis. However, Hydrostreamer requires users to provide runoff and stream network or catchment boundaries as inputs. It is not clear to me nonexperts can provide such information. Even if they can, there should be a minimal requirement to make sure that projects/coordinate systems used by these inputs are consistent with each. More descriptions on the pre-processing step are necessary.

2. The use of the interpolation methods implies that the resolutions of selected global runoff products shall be comparable to the catchment sizes of case studies. A threshold of watershed/catchment size should be provided so that the applicability of Hydrostreamer can be better understood.

3. It is not clear how the ancillary variables in dasymetric mapping are selected.

4. In the area-to-line interpolation method, it is assumed that contributing area can be replaced by the length of river segment. However, when the river network is delineated based on DEMs, it is typical to make an assumption on the threshold of stream cells. Such an assumption by itself could be subjective. Such uncertainty needs to be acknowledged.

5. The two routing methods are very simplified but can be reasonable options for watersheds of reasonable sizes. The instantaneous routing method is only applicable to large basins. Please add discussions on the size threshold. The constant velocity routing method is highly dependent on its parameter, the flow velocity. However, there is no discussion on how the parameter value is selected in the text.

6. Based on the case study presented, the tool can be useful for downscaling global runoff products at monthly scales or above. However, without validation in other flow regimes, it is hard to tell how transferable the results. Such a limitation needs to be acknowledged in the text.

7. In general, the style of writing needs to be improved to provide additional background materials for non-expert users.

8. The inputs/outputs of the case studies shall be provided for reproducibility.

**[GMDD](javascript:void(0))**

---

## Author Response (AR1)

**Response to reviewers and the edits made in the work**

Dear editor,

Please find below our responses to the review concerns we have posted to the interactive discussion, amended with the appropriate changes made in the manuscript. The manuscript has considerably improved following the suggestions of the reviewers, and we hope you and the reviewers find our revision satisfactory.

In addition to responding to the reviewers concerns, we've made a number of improvements to the code and added functionality, shown in Table 2 in Section 3.1 of the manuscript. These include e.g. the ability to compute hydrological signatures from the different timeseries relating to the hydrostreamer-object, or aggregating an attribute value for all upstream segments in the network. We wish users will find this additional functionality useful for their applications.

**Reviewer #1**

Dear Anonymous Reviewer #1,

Thank you for your remarks on our manuscript. Please find below our detailed responses to the two main points raised.

> **Comment 1.0:** In this manuscript, a useful tool is developed for non-hydrologist to use runoff products estimated by various land surface models. The tool mainly has three functions: 1) mapping the runoff components from the land surface model to units in the hydrological model, 2) modeling the river routing processes, and 3) assimilation via modeling averaging. The article is well written and well organized. I have a few suggestions which might help make the paper stronger. I think it could be done as a major revision.
>
> **C1.1**. In the manuscript two mapping methods are provided by the developed tool. The area-to-line interpolation is not looking reasonable to me. In this approach, the intersecting portion of the river line within the source zone is used as average weight.

**Response 1.1**: Thank you for your remark. We agree that our proposed Area-to-Line interpolation is unconventional and does not respect the actual drainage area delineation. Reduced data requirements may provide an advantage, , provided that certain conditions are met.

- The river network used for Area-to-Line interpolation needs to be sufficiently dense (with each source zone containing at least one river segment) in comparison to the input runoff data. We used 0.5° resolution input data with HydroSHEDS river network. The number of river segments in the source zones averages at 56 segments per source zone.
- Stream length does need to be a reasonable proxy for basin area. This is likely to be the case for sufficiently large basins with dense networks. We found that within our study area, the 3S basin, Pearson correlation between upstream river segment length and upstream basin

area is 0.998. We acknowledge that at an individual segment uncertainty is large, and therefore the method should only be used in basins where the area of source zones entirely contained in the basin is significantly larger than the area of partially covered source zones.

- Performance does need to be evaluated prior to use. The discharge estimates at the gauging stations used in our case study show that there is very little difference in goodness-of-fit statistics between Area-to-Line interpolation, DEM-delineated catchments, and Thiessen Polygon-based catchment estimation. This shows that the Area-to-Line interpolation can be used for catchments of size similar to the downstream stations in 3S (approximately 30 000 km$^2$) where the difference in performance between methods is nearly negligible (see Table A1). Results using the DEM-delineated catchments could be provided in the paper if necessary, but the Area-to-Line interpolation has been shown to have similar performance with lower input requirements.

These three points are emphasised in the revised manuscript, i.e. we explain why the good performance of the area-to-line interpolation in our case study does not necessarily transfer to every use case. Section 2.2.2 now includes the following: In our case study presented in Sect. 4, each source zone (a 0.5° grid; approximately 55 km at the equator) intersects on average 56 river segments. As the performance difference is small between Area-to-Line interpolation and area-based interpolation methods (Appendix A, Table A1), this can be considered a sufficient density for source zones in this resolution (but subject to case-by-case evaluation). Further, since the individual river segment length is not directly proportional to its individual catchment area, Area-to-Line interpolation should only be used for sufficiently large basins, where the area of source zones entirely contained in the basin is significantly larger than the area of partially covered source zones. Based on our case study, monitoring stations with a drainage area of at least 30 000 km2 show very small performance difference between Area-to-Line interpolation and area-based interpolation methods (Table A1). Due to the large uncertainty in runoff distribution to individual segments, we recommend that the suitability of Area-to-Line interpolation be performance-evaluated on a case-by-case basis.

**C1.2**: The approach did not respect the actual drainage area controlled by the river line, actually, somehow recalculate the drainage area of each river line during the mapping process.

**R1.2**: The Area-to-Line method assumes that the length of the river segment within a runoff source zone approximates the drainage area of that segment within the same zone. We acknowledge that this assumption leads to a poor redistribution of runoff at an individual segment level. However, as mentioned in our answer R1.1 to the previous point, the difference to the catchment-based methods is very small at the gauging stations. This happens because the runoff contribution falling within a certain basin changes only at the boundary of the basin, and thus as the basin size increases, the contribution of the boundary segments gets increasingly small compared to the contribution of segments in the interior of the basin.

This point is covered the same section 2.2.2 as our response to C1.1, emphasising the evaluation on a case-by-case basis.

**C1.3**: In an extreme case, if a river line flows along the boundary of the grid, then no runoff will contribute to that river line.

**R1.3**: Thank you for pointing this out. We tested whether this actually happens within our code, and found that in such case, a segment is assigned runoff from both source zones according to the length at the grid boundary. We have fixed this, and the contribution is now evenly split between grid cells intersecting the line at the boundary, as is appropriate.

**C1.4**: Two simple methods are offers for river routing. I would suggest adding more routing options for example diffusive wave, hydrological routing approaches. It would be interesting to also assimilate the simulated streamflow from different routing methods, not only using different runoff inputs.

**R1.4**: Thank you for your suggestion. We note that the primary emphasis of the paper is on the mapping of gridded runoff onto river networks, which can then also be used with existing routing software. The two simple routing methods included are already used in different global modelling efforts or applications using data from global model runs (see e.g. Telteu et al., 2021; Lehner and Grill, 2013; Munia et al., 2018). However, after consideration of your suggestion and the application of hydrostreamer to catchment scales, we decided to implement a more comprehensive Muskingum-Cunge routing algorithm. This improves the usability of hydrostreamer particularly for sub-monthly timescales. For monthly timeseries, there is little difference between the existing simple routing methods and the added Muskingum-Cunge routing algorithm at our study area, the 3S basin. In the revised manuscript, we now provide a short comparison of the outcome resulting from the three applied routing methods in the supplementary materials, referenced in the main text.

Section 2.3 now describes our approach to Muskingum-Cunge: The third routing option implemented in hydrostreamer is the Muskingum-Cunge routing algorithm (Cunge, 1969; Ponce, 2014). Muskingum-Cunge is a modified version of the original Muskingum routing method (Chow, 1959) where routing parameters *k* and *x* are derived from hydraulic data and does not require observation data to calibrate against. Full derivation and explanation of the Muskingum-Cunge routing can be found in Ponce (2014). The algorithm requires extensive user input in the form of river cross-sections (i.e. shape, channel width, flow depth), river bed roughness (Manning's roughness coefficient), and river bed slope, which are commonly available only for certain locations. Consistent with the desire to minimise data requirements, the hydrostreamer implementation of Muskingum-Cunge provides defaults and therefore requires the user only to provide main parameters: 1) Manning's roughness coefficient (for readers unfamiliar with Manning's coefficient, Arcement and Schneider (1989) provide an extensive guide on its estimation), 2) bed slope (precomputed bed slopes are available e.g. from the HydroATLAS (Linke et al., 2019) database which can be directly used in hydrostreamer), and 3) channel width. An estimate of the channel width can be computed using a power-law relationship (Leopold and Maddock Jr., 1953)

$$W = aQ_{ref}{}^{b} \qquad (15)$$

where $a$ and $b$ are parameters to be estimated and $Q_{ref}$ is the reference discharge, and $W$ is the channel width. Hydrostreamer has a built-in estimates for $a$, $b$ from Moody and Troutman (2002) and Allen et al. (1994). $Q_{ref}$ is estimated from the inflowing discharge timeseries for each river segment using Eq. 16,

$$Q_{ref} = \min(Q_{in}) + \frac{\max(Q_{in}) - \min(Q_{in})}{2} \qquad (16)$$

where $Q_{in}$ is the timeseries of discharge inflowing to the river segment. Alternatively, the user can provide their own parameters for each $a$, $b$, and $Q_{ref}$ Vatankhah and Easa (2013) derived a relationship between discharge $Q$ and flow area based on channel width. Their approach is used here to estimate flow depth assuming a rectangular river cross-section.

We have further added an Appendix B, which includes a comparison of the three routing methods for our case study.

**Reviewer #2**
* * *
Dear Anonymous Reviewer #2,

thank you for providing insightful comments for our manuscript. Please find below our detailed point-by-point response to the concerns raised.

> **C2.0**: The study presents an R package, a software library hydrostreamer v1.0 which aims to improve the usability of existing runoff products by addressing the Modifiable Area Unit Problem, and allows nonexperts with little knowledge of hydrology-specific modelling issues and methods to use them for their analyses. The topic is well suited for publication in GMD, however, the manuscript has some unclear reasoning that requires significant revision before the manuscript can be accepted for publication. My major comments are provided as follows.

**R2.0**: Thank you for your remark. We hope you find our response and amendments to the manuscript satisfactory.

> **C2.1:** The work was motivated by providing a tool that can be used by non-hydrologist to downscale global runoff products to river-basin scale for follow us analysis. However, Hydrostreamer requires users to provide runoff and stream network or catchment boundaries as inputs. It is not clear to me nonexperts can provide such information. Even if they can, there should be a minimal requirement to make sure that projects/coordinate systems used by these inputs are consistent with each. More descriptions on the pre-processing step are necessary.

**R2.1**: Thank you for this pointer. Indeed, the user does need a minimum level of GIS expertise and proficiency with R, which we previously took for granted. It does not need specialist hydrology expertise. Further, Hydrostreamer includes functions to evaluate performance and therefore support non-expert judgement about the adequacy of results.

We now make explicit our implicit assumptions about the user's expertise. We have added to the revised manuscript a section "2.1 Obtaining data and pre-processing".

To eliminate errors due to inconsistent coordinate systems, Hydrostreamer makes use of the *sf*-package (Pebesma, 2018) for geoprocessing, which does not allow the use (outputs an error) of input data with non-matching coordinate reference systems, and outputs a warning if a non-projected geographical coordinate system is used.

Due to the amount of amendments to the manuscript to cover this comment, we do not include the changes made here. However, the revised manuscript contains many more recommendations for non-experts, and all our changes can be tracked from the markup included in the revision.

**C2.2**: The use of the interpolation methods implies that the resolutions of selected global runoff products shall be comparable to the catchment sizes of case studies. A threshold of watershed/catchment size should be provided so that the applicability of Hydrostreamer can be better understood.

**R2.2**: Thank you for the remark. When the user provides catchments for the river segments, there is no implicit limit on the resolution of input. If the target zones are larger than runoff source zones, the method effectively upscales rather than downscales. If the target catchments are of much higher resolution than source zones the method performs downscaling (see Kallio et al., 2019). Area-to-Line interpolation, however, is critically dependent on proper scales, as the reviewer notes in comment number 4. This is because all source zones must contain at least one intersecting river segment in order for runoff to be assigned to the river segment. We discuss this in section 2.2.2 (see R1.1) and provide a recommendation based on our case study.

**C2.3**: It is not clear how the ancillary variables in dasymetric mapping are selected.

**R2.3**: In general, ancillary variables are selected based on their presumed or tested relationship with the spatial distribution of the variable of interest.

We have now made the choice of ancillary variables clearer, and provide examples about possible sources for ancillary data.

We amended Section 2.2.1 with the following passage: The dasymetric variable(s) should be selected such that it (they) describe the distribution of runoff *within* each source zone. Potential variables include topographic information (elevation, topographic indices; the case study in this paper uses a topographic index as a dasymetric variable), landuse, soil type, climate information (precipitation, temperature, evapotranspiration), and so on. The choice depends on the availability of data for each individual target zone as well as on the hydrological understanding of the user.

       **C2.4**: In the area-to-line interpolation method, it is assumed that contributing area can be replaced by the length of river segment. However, when the river network is delineated based on DEMs, it is typical to make an assumption on the threshold of stream cells. Such an assumption by itself could be subjective. Such uncertainty needs to be acknowledged.

**R2.4**: We agree, such a decision is indeed subjective. In an ideal case the user would be able to use a river network product built with an appropriate choice of a threshold. Since a non-expert may not be aware of such delineation techniques, we have added an explicit mention of this, as you suggest, to the new section on data and preprocessing. See also the responses to Reviewer 1 (R1.1) regarding the conditions required for area-to-line interpolation to be applicable.

To address this, the new Section 2.1 covers this as follows: Hydrostreamer further provides an optional auxiliary function *create_river()* which can be used to extract a river network and catchment areas from a DEM for each river segment. The function requires an external program, SAGA GIS (Conrad et al., 2015) to be installed, and requires definition of a threshold for the size of the stream (the Strahler stream order) at which point river line extraction starts. The selection of the threshold should be guided by the resolution of the source zones as well as understanding of the hydrology within the basin. We recommend visual inspection of the extracted river network as well as their corresponding catchment areas.

       **C2.5**: The two routing methods are very simplified but can be reasonable options for watersheds of reasonable sizes. The instantaneous routing method is only applicable to large basins. Please add discussions on the size threshold. The constant velocity routing method is highly dependent on its parameter, the flow velocity. However, there is no discussion on how the parameter value is selected in the text.

**R2.5**: We agree that instantaneous routing is limited – not just by basin size, but also temporal resolution of output. Given that flow timing can also depend on shape of a basin, it is not possible to define fixed thresholds and evaluation of performance is instead recommended. This is now emphasised in the revised manuscript. To help the user, we have also added functionality to the package to evaluate the applicability of instantaneous routing, as follows: However, it assumes that all runoff generated at a timestep *t* will drain through the entire river network within that same timestep. The applicability of this assumption is therefore limited to catchments where the timestep length far exceeds the maximum river network length. One can evaluate the applicability of the instantaneous routing (which in fact takes one timestep) using Eq. (9):

$$M = \frac{\max L_{up}}{V_{max}} \frac{1}{s} \tag{9}$$

where $M$ is a dimensionless ratio between the time it takes for water to flow through the maximum upstream length of the river system $L_{up}$ (in meters) at a maximum realistic average flow velocity $V_{max}$ (default 1 meter per second) during a timestep of length $s$ (in seconds). $M$ can be interpreted so that, for example, when $M = 0.1$, 10% of the runoff generated at the most distant upstream location does not flow through the outlet within a single timestep. The evaluation can be carried out

using the function *evaluate_instant_routing()*. If $M$ is found to be too large for the application, the constant flow velocity or Muskingum-Cunge option may be more appropriate.

Considering your as well as the other reviewer's comments (C1.2) on the routing, we have added an implementation of Muskingum-Cunge routing algorithm to Hydrostreamer which should increase the utility of the package for smaller basins as well as for higher temporal resolutions than a month. Please find the added description of the routing method in response R1.2.

We have made the section on routing clearer on the assumptions made. The flow velocity in this case study is the same 1 m s$^{-1}$ as adopted by in HYDROROUT and LPJmL (Telteu et al., 2021). That is, in addition to using data from global runoff models, a hydrostreamer user can use the default flow velocity from those models in absence of better information, but can use spatially varying information if available. The paragraph describing constant velocity routing now includes the following sentence: The default flow velocity of 1 meter per second is adopted e.g. in HydroROUT and LPJmL (Telteu et al., 2021).

**C2.6**: Based on the case study presented, the tool can be useful for downscaling global runoff products at monthly scales or above. However, without validation in other flow regimes, it is hard to tell how transferable the results. Such a limitation needs to be acknowledged in the text.

**R2.6**: Thank you for this remark. Indeed, our experiment is made on a monthly scale, for which the instantaneous and constant-velocity routing solutions are adequate, given that a number of global hydrological models or applications using their outputs use similar routing schemes (see e.g. Telteu et al., 2021; Lehner and Grill, 2013; Munia et al., 2018). However, we recognize these are not necessarily adequate for smaller basins or for shorter timesteps. We have added a short comparison of the three routing methods as a new Appendix B, showing that while there is a difference, it is practically negligible.

**C2.7**: In general, the style of writing needs to be improved to provide additional background materials for non-expert users.

**R2.7**: We have revised the text, and believe it is now more approachable to non-expert users. Particularly, we have added much more description on potential data sources, and added many recommendations on the key steps in hydrostreamer application.

**C2.8**: The inputs/outputs of the case studies shall be provided for reproducibility.

**R2.8**: The code and input data are provided in a Zenodo-repository (https://zenodo.org/record/3987723). This repository contains all data except for discharge observations, which we are unable to provide openly due to the licence. Thus, the code can be run until the model averaging steps without observation data.

We are updating the repository with the output runoff and discharge for all 2115 river segments for the study period 1980-2010. The updated repository can be found at
https://zenodo.org/record/4739212

**References**

Kallio, M., Virkki, V., Guillaume, J. H. A., and van Dijk, A. I. J. M.: Downscaling runoff products using areal interpolation: a combined pycnophylactic-dasymetric method, in: El Sawah, S. (ed.) MODSIM2019, 23rd International Congress on Modelling and Simulation., 23rd International Congress on Modelling and Simulation (MODSIM2019), https://doi.org/10.36334/modsim.2019.K8.kallio, 2019.

Lehner, B. and Grill, G.: Global river hydrography and network routing: baseline data and new approaches to study the world's large river systems, Hydrol. Process., 27, 2171–2186, https://doi.org/10.1002/hyp.9740, 2013.

Munia, H. A., Guillaume, J. H. A., Mirumachi, N., Wada, Y., and Kummu, M.: How downstream sub-basins depend on upstream inflows to avoid scarcity: typology and global analysis of transboundary rivers, Hydrol. Earth Syst. Sci., 22, 2795–2809, https://doi.org/10.5194/hess-22-2795-2018, 2018.

Pebesma, E.: Simple Features for R: Standardized Support for Spatial Vector Data, R J., 2018.

Telteu, C.-E., Müller Schmied, H., Thiery, W., Leng, G., Burek, P., Liu, X., Boulange, J. E. S., Seaby Andersen, L., Grillakis, M., Gosling, S. N., Satoh, Y., Rakovec, O., Stacke, T., Chang, J., Wanders, N., Shah, H. L., Trautmann, T., Mao, G., Hanasaki, N., Koutroulis, A., Pokhrel, Y., Samaniego, L., Wada, Y., Mishra, V., Liu, J., Döll, P., Zhao, F., Gädeke, A., Rabin, S., and Herz, F.: Understanding each other's models: a standard representation of global water models to support improvement, intercomparison, and communication, Geosci. Model Dev. Discuss., 1–56, https://doi.org/10.5194/gmd-2020-367, 2021.

---

## Author Response (AR2)

**Response to the editor**

Dear Dr. Mills,

Thank you very much for the positive feedback to our reviewed manuscript. We have made the further amendments to the manuscript, as suggested in your review report, explained below.

> \* In C1.3, the reviewer noted the edge case in which a river line flowing along the boundary of the grid will result in no runoff contribution to that river line. The authors note in R1.3 that they tested this possibility and fixed it, and that "the contribution is now evenly split between grid cells intersecting the line at the boundary". I would like to see the authors explicitly state in the text that this case is handled in this way: In their code they have fixed the reviewer's concern, but I do not see (please tell me if I have overlooked it) where the text explains this. If reviewer 1 was led by the text to conclude that there is a problem with river lines along the boundary of the grid, I believe that other readers may also wonder about this if this is not explicitly addressed in the text.

We have added the following short explanation in page 10, section about Area-to-Line interpolation, right under Equation 4.

In some combinations of river lines and source zones, the river may flow exactly along the boundary of two or more source zones. Since this portion intersects both source zones, such cases are explicitly handled by hydrostreamer to split the contribution evenly among the source zones for the portion of river line at the boundary.

> \* I do not see where reviewer comment C2.6 about flow regimes below monthly scales has been explicitly addressed in the text. The Appendix B mentioned in R2.6 is a valuable addition to the paper. However, I would like to see a sentence or two explicitly mentioning the need for caution around the limitation mentioned in C2.6. I actually believe that the authors have already written something appropriate in R2.6; they just need to put this in the paper. (Again, let me know if I have overlooked this.)

In page 14, to the end of the river routing section, we added a short paragraph mentioning validation issue about the temporal resolution.

Note that, our case study example and Appendix B provide validation for the routing with monthly timeseries only. We therefore recommend caution and careful review of hydrostreamer outputs in applications using sub-monthly timeseries, until proper validation for the method is published.

We hope these amendments cover the two points we had not explicitly included in the manuscript. The only other change made to the manuscript at this stage is that we have unified the spelling of Area-to-Line interpolation to have upper case first letters. The previous version had mixed capital and lower and upper case letters.